# A hybrid method for reconstructing the historical evolution of aerosol optical depth from sunshine duration measurements

William Wandji Nyamsi[1], Antti Lipponen[1], Arturo Sanchez–Lorenzo[2], Martin Wild[3], and Antti Arola[1]

[1]Finnish Meteorological Institute, Kuopio, Finland
[2]Department of Physics, University of Extremadura, Badajoz, Spain
[3]ETH Zurich, Institute for Atmospheric and Climate Science, Zurich, Switzerland

*Correspondence to*: William Wandji Nyamsi (william.wandji@fmi.fi)

**Abstract.** A novel method has been developed to estimate aerosol optical depth (AOD) from sunshine duration (SD) measurements under cloud-free conditions. It is a physically-based method serving for the reconstruction of the historical evolution of AOD during the last century. In addition to sunshine duration data, it requires daily water vapor and ozone products as inputs taken from the ECMWF twentieth century reanalysis ERA-20C, available at global scale over the period 1900–2010. Surface synoptic cloud observations are used to identify cloud-free days. For sixteen sites over Europe, the accuracy of the estimated daily AOD, and its seasonal variability, is similar or better than those from two earlier methods when compared to AErosol RObotic NETwork measurements. In addition, it also improves the detection of the signal from massive aerosol events such as important volcanic eruptions (e.g., Arenal and Fernandina Island in 1968, Chichón in 1982 and Pinatubo in 1992). Finally, the reconstructed AOD time series are in good agreement with the dimming/brightening phenomenon and also provides preliminary evidence of the early-brightening phenomenon.

## 1. Introduction

Aerosols in the atmosphere are generally produced by natural and anthropogenic mechanisms, e.g. dust and sea salt triggered by wind-driven processes or carbonaceous aerosols (organic and black carbon) from combustion in urban/industrial processes or from biomass burning. They play a crucial role in the Earth's climate through their direct effects by scattering and absorbing solar radiation (Charlson et al., 1992; Hansen et al., 1997) and their indirect effects by acting as cloud condensation nuclei (Tang et al., 2016).

In the IPCC Fifth Assessment Report (IPCC, 2013), aerosols are mentioned as the largest contributor to the large uncertainties in the projections of climate change. A significant source in this uncertainty is linked to the limited knowledge of the historical evolution of aerosol load. In addition, the role played by aerosols in the dimming and brightening phenomenon is not yet well–established (Wild, 2009, 2016). For that reason, it has become of great importance to the scientific community to estimate the historical evolution of aerosol load accurately.

Aerosol optical depth (AOD) has been widely used to represent the aerosol radiative impacts. It has been mainly measured using reference instruments, sun photometers for instance, from various ground-based networks. Among them are the Aerosol Robotic Network (AERONET; Holben et al., 1998), the Global Atmospheric Watch Precision Filter Radiometer network (McArthur et al., 2003) and the SKYradiometer NETwork (Aoki et al., 2006). The most widely used ground-based network is AERONET. Although AERONET already contains over 700 stations globally with a fairly good spatial coverage over land compared to many other observational networks, it still lacks of temporal coverage. AERONET has provided aerosol optical properties and AOD only since the 1990s at some sites, while the number of measurement sites started to substantially increase only in the early 2000s. The earliest records of satellite-based AOD are provided by TOMS (Total Ozone Mapping Spectrometer, e.g. Torres et al., 2002) and AVHRR (Advanced Very High Resolution Radiometer, Geogdzhayev et al., 2005) from 1979 and 1983 onwards, respectively. It is apparent that neither sun photometer nor satellite records of AOD are available for a long time period, which would extend before the 1980s. The pyranometer measurements of surface solar irradiance (SSI) are also very valuable to infer AOD (Lindfors et al., 2013; Huttunen et al., 2016). However, this type of measurements started to become available mainly only after the 1950s, with the establishment of numerous radiation sites during the International Geophysical Year (IGY) 1957–1958.

To overcome this scarcity of information on the evolution of AOD, especially before 1950, some researchers have used proxy-approaches. Thus, some of them have used sunshine duration (SD) measurements to infer AOD under cloud–free conditions (Sanchez–Romero et al., 2016; Li et al., 2016; Dumitrescu et al., 2017), because SD measurements offer remarkably long time series going beyond 1950 and with a noticeable worldwide spatial coverage. SD for a given period, mostly a day, is defined as the sum of the sub-periods for which broadband direct normal irradiance (DNI) exceeds the threshold value of 120 W m$^{-2}$ (WMO, 2008). This value is assumed to be the "burning threshold". From this definition, the evidence of the link between SD and AOD can be summarized as follows: an increase in AOD would decrease DNI yielding then less sub-periods when DNI would be greater than the burning threshold and therefore results in a reduction in SD. For a review of the topic, we refer to Sanchez–Romero et al. (2014).

The Campbell-Stokes sunshine recorder (CSSR) is one of the devices used to measure SD. It has been manufactured since the late 19$^{th}$ century to record the duration of the sunbeam through a burned trace on an appropriate card (Sanchez-Lorenzo et al., 2013). The measurement of the length of the burned trace for a given card over the day gives the daily SD. Recently, automatic SD recorders have been developed and they are becoming more and more spatially distributed (Wood et al., 2003; Kerr and Tabony, 2004; Matuszko et al., 2015). These instruments are much more accurate than CSSR because they measure the beam irradiance over the day and thus count efficiently the duration during which the beam irradiance exceeds the 120 W m$^{-2}$ threshold. However, the measurement time series from the automatic SD recorders are too recent to provide long enough SD time series for our purpose. Thus, this study will mostly deal with measurements operated with CSSR.

There are two different previous approaches that have been published on the retrieval of daily AOD from daily SD measurements. These are the methods described in Sanchez–Romero et al. (2016) and Li et al. (2016). In the first approach, the central assumption is the station–by–station fitted linear relationship between SD fraction (SDF), i.e. SD normalized by the length of day from sunrise to sunset, and AOD measurements. Sanchez–Romero et al. (2016) applied the approach to SD stations throughout Spain and collocated AERONET stations. A similar approach was applied by Dumitrescu et al. (2017) over 57 Romanian stations. Both studies were limited to the summer season only to ensure a sufficient number of clear-sky days and thus a sufficient number of data pairs in the linear fit between SDF and AOD.

The second approach is a physically-based method, in essence similar to the direct sun methods applied with sunphotometer measurements to retrieve AOD, in which individual attenuators such as Rayleigh scattering, mixed gases, ozone and water vapor are removed from the overall attenuation. And the residual attenuation is then due to aerosols. This approach has the advantage that it can be applied at any time, i.e. does not depend on the season as the first approach, which needs enough collocated SDF and AOD measurements for the linear fit. Li et al. (2016) applied their approach over China with corrected SD measurements. Nevertheless, there was an overestimation in the retrieved monthly AOD when compared to AERONET measurements. This overestimation was due to an inadequate constant value used for the burning threshold, as we discuss below in more detail.

We propose a new and more accurate method for estimating AOD from SD measurements under cloud-free days. In a sense, it combines the best aspects from both Li et al. (2016) and Sanchez–Romero et al. (2016) with further enhancements in some parts. Among the novelties, the proposed method exploits a more accurate broadband DNI model and takes into account local conditions affecting SD measurements. Then, the proposed method is used to provide a realistic historical evolution of AOD over a few European stations.

The European Climate Assessment & Dataset (ECA&D) project has been selected for such long time series of SD measurements. About a hundred stations in Europe have collected SD measurements together with cloud cover data and other meteorological parameters such as air temperature and relative humidity. Few of them are collocated with AERONET stations providing AOD.

The article is organized as follows. In Section 2, a description of all data used in this study is given. Then, the two published state-of-the-art methods and the new hybrid method are described in Section 3. In Section 4, a detailed theoretical study is carried out on the relationship between AOD and SD with radiative transfer simulations and the influence of water vapor as well as ozone. In Section 5, the methodology of this study is presented. An assessment of clear sky day selection algorithm based on 6-hourly *vs* 1-hourly cloud cover measurements is carried out. The performance of the proposed method is compared to the two approaches. The results of different AOD estimates are seasonally inter-compared, analyzed and discussed in Section 6.

## 2. Data used in this study

### 2.1. AERONET measurements

AERONET is a well–known and globally distributed network of ground–based stations which are equipped with sun photometers measuring direct sun radiances at eight wavelength bands centered at 340, 380, 440, 500, 670, 870, 940, 1020 nm. The measurements are made every 15 minutes between sunrise and sunset with an accuracy of 0.01–0.02 (Eck et al., 1999). They are performed only under cloud–free conditions (Smirnov et al., 2000). From the AERONET direct sun measurements, daily AOD and total water column (TWC) are provided (Dubovik et al., 2000). In this paper, level 2.0 data are selected for their assured quality. The data can be downloaded from the website https://aeronet.gsfc.nasa.gov/new_web/download_all_v3_aod.html.

### 2.2. ECA&D database

Ground-based SD measurements and cloud information represented by total cloud cover (TCC) constitute useful data from this database. The data are from the time series of stations located over Europe. They are collected in the European Climate Assessment & Dataset project (https://www.ecad.eu). Both SD and TCC are available for about hundred stations, respectively. Since the data are provided at various temporal resolutions, daily SD and mean daily TCC within the calendar day are selected.

### 2.3. ECMWF total water vapor and ozone column

Among other products, the European Centre for Medium-range Weather Forecasts (ECMWF) provides reanalysis variables of TWC and total ozone column (TOC). These variables take into account several analyses of ground-based observations used as inputs of ECMWF model. They are considered as representing the atmospheric state fairly well. They are inputs for the physically–based method in order to remove the attenuation due to water vapor and ozone from broadband DNI.

In this article, we select mean daily TWC and TOC from the ECMWF twentieth century reanalysis ERA-20C covering the period 1900–2010 (Poli et al.; 2016). A comparison between ECWMF and AERONET daily TWC is carried out over Europe with AERONET measurements serving as reference. On average, we found that the ECMWF values slightly underestimate TWC by 4%, or below 1 kg m$^{-2}$ with a

correlation coefficient of 0.8 and a small standard deviation. This result demonstrates the good accuracy of the ECMWF reanalysis products. Because of their extended temporal coverage, the ECMWF products are mainly used for the reconstruction of the historical evolution of aerosol load and have been downloaded from the website https://www.ecmwf.int/en/forecasts/datasets/reanalysis-datasets/era-20c

## 2.4.    OMI TOC

The Ozone Monitoring Instrument (OMI) on board the NASA EOS Aura spacecraft provides TOC measurements since August 9, 2004. Several studies have demonstrated the quality of TOC estimates when compared to ground-based measurements. Daily OMI TOCs are available from the website

10    https://acdisc.gesdisc.eosdis.nasa.gov/data/Aura_OMI_Level3/OMTO3d.003/ since October 1, 2004. In this study, OMI-measured TOC is used from October 1, 2004 onwards while ECMWF TOC is used for October 1, 2004 backwards mostly for the historical reconstruction. Our comparisons of TOC from ECMWF and OMI serving as reference show on average a high correlation of 0.9, a low bias of +3% and very limited spread of points denoting the accuracy of ECMWF TOC.

## 2.5.    MODIS MxD08_D3 products

The MODIS MxD08_D3 products, namely MOD08_D3 and MYD08_D3 data collected from the Terra and Aqua platforms, respectively, are daily level-3 satellite atmosphere datasets based on NASA's Moderate Resolution Imaging Spectroradiometer (MODIS) observations. MODIS instruments are flying

20    onboard both Terra (morning overpass) and Aqua (afternoon overpass) satellites and for aerosols, they provide information for clear-sky pixels over both land and ocean. Despite of the coarser spatial resolution than the corresponding one for level-2 product, the level-3 product was used because it takes into account all overpasses of the instrument during the same day making it to be more representative of daily average aerosol properties than the level-2 instantaneous data products. In this level-3 product, the

25    aerosol retrievals are spatiotemporally aggregated into a daily dataset with spatial resolution of 1° latitude x 1° longitude. These aerosol data cover the period of early 2000 (Terra) and mid-2002 (Aqua) onwards. We use the AOD product at 550 nm that is a combined product of Collection 6.1 Dark Target (Levy et al.,        2013)        and        Deep        Blue        algorithms        (Hsu        et        al.,        2013) (AOD_550_Dark_Target_Deep_Blue_Combined_Mean). Over land, we use the Ångström exponent

data from Deep Blue (Deep_Blue_Angstrom_Exponent_Land_Mean, 412-470 nm) that is the only Ångström exponent data given in the dataset. MODIS data are downloadable for example from the Level-1 and Atmosphere Archive & Distribution System (LAADS) Distributed Active Archive Center (DAAC) at https://ladsweb.modaps.eosdis.nasa.gov/. We assume that the Aqua aerosol data are representative of the daily AOD estimates. If there is no Aqua AOD available, the Terra AOD is used instead.

## 2.6.  MERRA-2 reanalysis

The Modern-Era Retrospective analysis for Research and Applications, Version 2 (MERRA-2) is a global meteorological reanalysis developed by NASA's Global Modeling and Assimilation Office (Gelaro et al., 2017). This global dataset assimilates several satellite and ground-based measurements and regarding aerosol products, there are, for example, ground–based measurements from AERONET, MODIS observations from both Terra and Aqua satellites, Multi-angle Imaging SpectroRadiometer (MISR) data from Terra satellite, and Advanced Very-High-Resolution Radiometer (AVHRR) from NOAA Polar Operational Environmental satellites (Randles et al., 2017). MERRA-2 data cover the period from 1980 to the present and the spatial resolution is 0.5° latitude x 0.625° longitude (Molod et al., 2015). In this study, we use the hourly data of total aerosol extinction AOD (TOTEXTTAU) at 550 nm and the total aerosol Ångström Exponent (470-870 nm, TOTANGSTR) available through the Goddard Earth Sciences Data and Information Services Center (GES DISC; http://disc.sci.gsfc.nasa.gov/mdisc/). All hourly data between sunset and sunrise are averaged in order to retrieve the daily MERRA-2 estimates.

## 3.  Description of the methods

## 3.1.  Linear regression method (LRM)

In order to retrieve AOD from SD observations, Sanchez–Romero et al. (2016) first computed the slope $a$ and the intercept $b$ of the linear regression between daily SDF and daily AOD at 440 nm under cloud-free conditions for each collocated AERONET and SD stations as follows:

$$AOD = a\,SDF + b \tag{1}$$

A day is considered as cloud-free if the average of three-daily observations is rounded to 0 okta. For a given station, the linear fit between SDF and AOD is accepted if the correlation coefficient is statistically significant with respect to a maximum $p$-value of 0.05.

 ### 3.2. Physically–based method (PBM)

The concept of this method is based on the simplified broadband DNI models. Let $G_b$ denote the broadband DNI received on a plane normal to the Sun rays at ground level, it is generally defined as:

$$G_b = \varepsilon \, G_o \, T_R \, T_g \, T_o \, T_w \, T_a \tag{2}$$

where $\varepsilon$ is the Sun–Earth distance correction factor depending on the day of the year, $G_o$ is extra-terrestrial irradiance received on a plane normal to the Sun rays also known as solar constant which is 1367 W m$^{-2}$, $T_R, T_g, T_o, T_w, T_a$ are individual broadband transmittances of the main attenuators i.e. Rayleigh scattering, uniformly mixed gases, ozone absorption, water vapor absorption and aerosol extinction, respectively. If $G_b$ and attenuator transmittances are known, the broadband aerosol transmittance can be computed as follows:

$$T_a = \frac{G_b}{\varepsilon \, G_o \, T_R \, T_g \, T_o \, T_w} \tag{3}$$

The broadband aerosol transmittance is mathematically defined as:

$$T_a = \exp(- \, m_a \, BAOD) \tag{4}$$

where $m_a$ is the optical aerosol mass and $BAOD$ is broadband AOD. Several researchers have shown that BAOD can be considered as approximately equal to AOD at 750 nm (Qiu, 1998; Molineaux et al., 1998).

To retrieve AOD from SD observations, Li et al. (2016) assumed the diurnal uniformity of AOD, water vapor, ozone from dawn to dusk. They used the simplified broadband DNI model developed by Paulescu and Schlett (2003) for estimating broadband transmittance of each attenuator with a common optical air mass adopted for all attenuators as defined by Kasten and Young (1989). In Eq. (2), $G_b$ is set to the burning threshold of 120 W m$^{-2}$ according to WMO (2008). The implication of diurnal uniformity of SD

is that SD can be converted to hour angle ($\omega$) i.e. $\omega = 15(\frac{SD}{2})$ in degrees. From an hour angle, in turn, when the latitude and the solar declination angle are known, the solar zenith angle ($\Theta_s$) can be computed. Then, the solar zenith angle is used to compute transmittance and air mass. With this approach, AOD is estimated at the instant close to sunrise/sunset under a fully cloud-free day, at an instant when the burned trace becomes visible/extinct. For more detailed information on this method, please refer to Li and al. (2016).

In addition, Li et al. (2016) applied a correction to the SD data. They added a small constant of a few percent to the reported SD data, due to the systematic bias found when compared retrieved AOD to MODIS-AOD. Nevertheless, an overestimation remained in the monthly mean comparisons between retrieved AOD and AERONET measurements. This overestimation was due to an unsuitable constant value for the burning threshold, as we discuss below in more detail.

### 3.3.    New Hybrid method (NHM)

One clear advantage of the Li et al. (2016) method is that it does not require, in principle, ancillary AOD measurements for the training as the method by Sanchez–Romero et al. (2016). Moreover, as a physically-based approach, it is an attractive option to estimate AOD from SD measurements. However, we found two points where this approach can be improved. They can be summarized as follows: (1) we make use of more accurate broadband transmittances for each atmospheric attenuator and (2) we establish seasonal station specific burning thresholds, since this threshold is clearly varying as a function of season, station and instrument conditions. AOD information is needed for the second improvement. Therefore, we exploit prior ground-based AOD measurements such as AERONET measurements. If not available, we can exploit satellite-based AOD from MERRA-2 because of its complete spatial coverage. Because, the method uses AERONET measurements as in Sanchez–Romero et al. (2016), we call our new method a hybrid method (NHM).

### 3.3.1.  A more accurate broadband DNI model

Since for cloudless days the first/last burned traces occur close to sunrise/sunset, it is crucial to select a broadband DNI model, which performs well at high SZAs. In the literature, several broadband DNI

models can be found, which perform well when all SZAs are included, but very few of them are accurate at high SZAs.

The broadband model can be considered as accurate enough if the broadband transmittance of each relevant attenuator is also accurate enough. One limitation in the Li et al. (2016) approach is the use of a common optical mass for all attenuators. Gueymard (2003a) compared different formulations for optical mass of each attenuator, as used in several broadband DNI models, to the optical mass computed from the radiative transfer model (RTM) SMARTS version 2.9.2 (Simple Model of the Atmospheric Radiative Transfer of Sunshine, Gueymard, 1995) serving as reference. Gueymard (2003a) demonstrated that the differences are significant at high SZA for ozone, water vapor and aerosol contents.

Gueymard (2003a) also compared broadband transmittances for each attenuator of these different broadband models against the corresponding broadband transmittance from SMARTS and found significant differences between models. From this study, the most accurate models for each attenuator can be identified as did by Gueymard (2003b). The Paulescu et al. (2003) model was not included in the 21 investigated models because that model was not yet published when the Gueymard (2003a) carried out the study.

The broadband DNI model applied in our method here is a collection of the most accurate models for broadband transmittance and optical mass of each attenuator. All necessary equations are explicitly given in the Appendix. The performance of the model has been assessed with respect to its capability to estimate accurately DNI for all SZAs and especially at high SZA. Results from the new DNI model and the Paulescu et al. (2003) model were compared to results from libRadtran (Emde et al., 2016; Mayer and Kylling, 2005) serving as reference.

A set of 12, 000 clear sky atmospheric states was built by means of the Monte-Carlo technique. Each state is a combination of eight variables described as follows: atmospheric profiles from Air Force Geophysics Laboratory standards, SZA, TOC, TWC, elevation of the ground above sea level, AOD at 1000 nm, Ångström exponent and aerosol type. The value of each variable was randomly selected by taking into account their modelled marginal distribution established from observations following Table 2 of Wandji Nyamsi et al. (2017) except that SZA varies between 75° and 89.9°. For all radiative transfer simulations, a pseudo-spherical atmosphere was assumed accounting for the sphericity of the atmosphere necessary to accurately compute optical masses at high SZA. In addition, the most improved version (katoandwandji included in libRadtran, Wandji Nyamsi et al., 2014; 2015) of spectral resolution of Kato

et al. (1999) was selected for band parameterization to calculate the broadband shortwave irradiance because several studies have demonstrated the accuracy of its results when compared to line-by-line calculations.

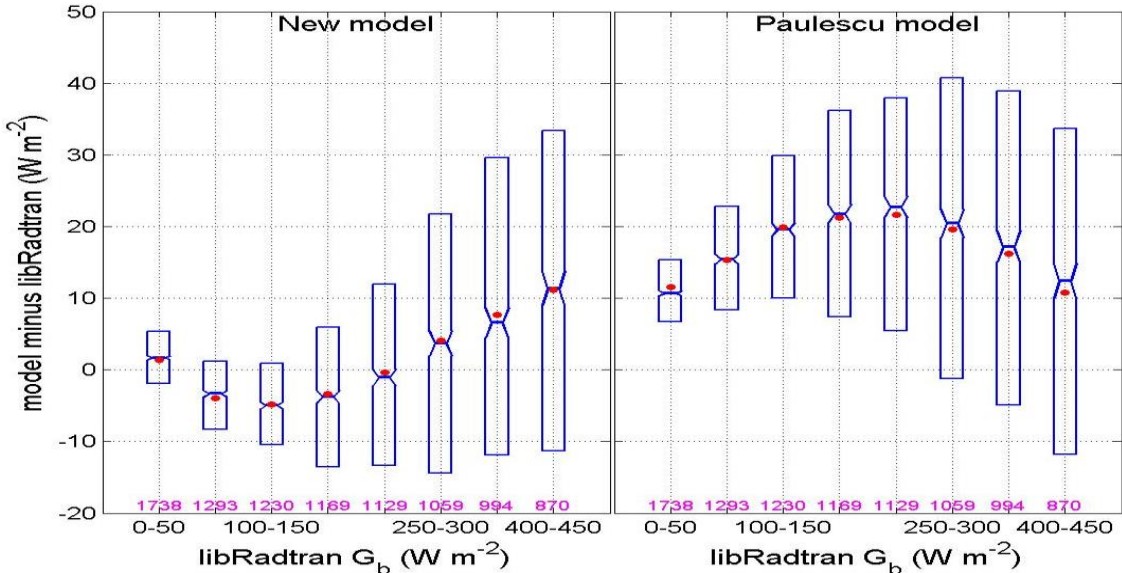

**Figure 1**: Dependence of deviations between estimated and libRadtran $G_b$ for the new and Paulescu models. The red dot indicates the mean, the limits of the boxes are $1^{st}$, $2^{nd}$ (median), $3^{rd}$ quartiles. The pink number is the number of data in a DNI range

Figure 1 displays the difference between estimated and libRadtran simulated values as function of
libRadtran DNI range for both models. It shows that there is a clear overestimation by the Paulescu model. Especially close to WMO (2008) burning threshold i.e. 120 W m$^{-2}$, there is noticeable overestimation from Paulescu when compared to the new model results where the bias is much smaller. This demonstrates the accuracy of the new model and also the fact that the use of the Paulescu model has partially contributed to the errors found in the Li et al. (2016) approach when deriving AOD from SD
measurements.

### 3.3.2. Seasonal variability of burning threshold

Despite of the recent progress made for accurate SD measurements by means of automatic SD recorder, most of the historical SD measurements have been recorded with traditional devices such as the CSSR
or Jordan instruments. Since our objective is to estimate AOD over a period as long as possible, using

data from Europe in our validation, CSSR is the instrument of our interest. The observational errors with CSSR are various. These have been discussed in the literature (Jaenicke and Kasten, 1978; Helmes and Jaenicke, 1984, Sanchez-Romero et al., 2015). In the following paragraph, a detailed discussion is provided.

Among the sources of errors, three situations can be listed as follows. First, if there is water deposit and dew on the glass sphere especially in the morning, more energy or irradiance is needed to warm the glass sphere to be able to focus the sunrays on the card for producing the burned trace (Painter et al., 1981). Second, in case there is moisture in the card, it needs to be dried enough to make the burned trace visible, implying also the need for additional energy. Third, there is variability in the card types with different
properties, for instance, absorption of water by the card (Helmes and Jaenicke, 1984). The errors with the CSSR are also caused by the local effects of the temperature and relative humidity on the burn card (Ikeda et al, 1986).

A need to use seasonal burning threshold was previously reported (Bider, 1958; Jaenicke and Kasten, 1978; Baumgartner, 1979; Painter et al., 1981). Based on several analyses, it has been found that the
burning threshold could range from 70 to 385 W m$^{-2}$. Painter et al. (1981) mentioned that some daily thresholds could reach values as high as 400 W m$^{-2}$ under some particular atmospheric conditions. The threshold is typically higher in wintertime than in summertime due to the variability in temperature and relative humidity. Therefore, there is a clear evidence for a seasonal variability of burning thresholds in CSSR measurements depending on the local conditions at the given station (Sanchez-Romero et al.,
2015).

The concept of this proposed method is to determine local monthly averaged burning thresholds and then use these thresholds in the PBM to derive BAOD as follows, by combining equations (3) and (4):

$$BAOD = - \frac{\ln(\frac{\overline{G_b}(mm)}{\varepsilon \, G_o \, T_R \, T_g \, T_o \, T_w})}{m_a} \qquad (5)$$

where $mm$ is a month between January and December, $\overline{G_b}(mm)$ is the computed effective monthly
burning threshold. This latter takes into account as many factors as possible affecting SD measurements. It is computed by using the new model with appropriate atmospheric inputs.

## 4. On the relationship between AOD and SDF

As explained above, the effective burning threshold typically exhibits a comprehensive variability from wintertime to summertime. We investigated the nature of relationship between AOD and SDF for various thresholds for a given site located, for instance, at a latitude of 35 N used for SZA computations ranging from 80° and 88°. In this case, a typical atmospheric state is considered here: TOC of 350 DU, elevation of 500 m, Ångström exponent of 1.3, TWV of 3 cm. This given atmospheric state is then associated with each aerosol content from a set of 1000 AODs at 550 nm built following the Monte Carlo draws as described previously. This yields 1000 realistic atmospheric states.

For each atmospheric state, DNI is simulated from sunrise to sunset with a high temporal resolution. For a given burning threshold, the length of the period for which DNI is greater than the burning threshold is derived. Then, it is converted to SDF. Figure 2 shows the curve of the change of AOD at 750 nm with SDF for three burning thresholds: 120 W m$^{-2}$, 250 W m$^{-2}$ and 400 W m$^{-2}$. According to WMO (2008), the typical range of SDF under cloud-free conditions is between 0.7 and 1. Regardless of the burning threshold value, the relationship tends to be linear for AOD greater than 0.1 and exhibits a non-linearity at AOD values below 0.1 (Fig. 2).

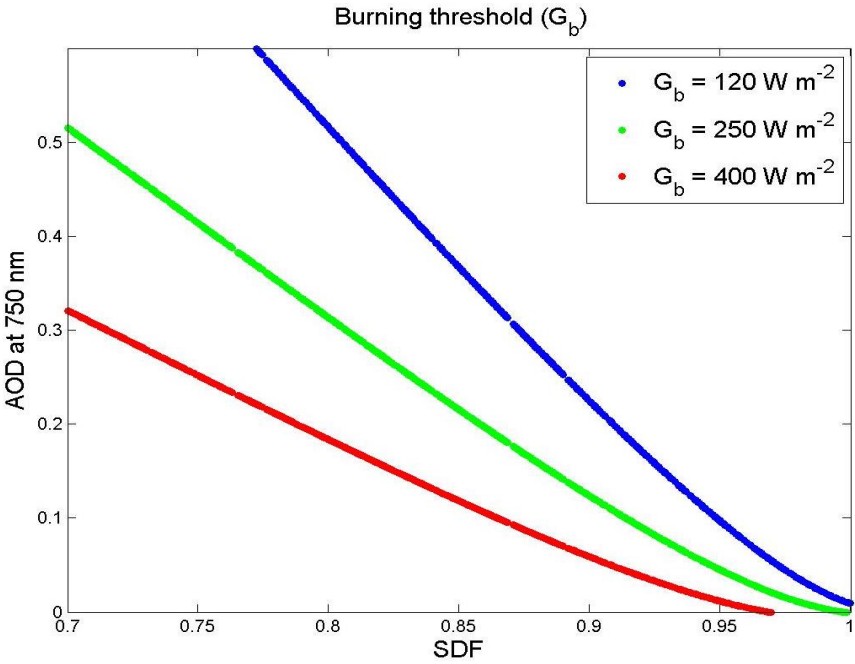

**Figure 2**: AOD at 750 nm as a function of SDF for three burning thresholds.

Influences of water vapor content and then ozone content are analyzed. For doing so, the previous set of atmospheric states is used twice: one with integer values of TWV between 1 cm and 7 cm instead of a fixed value of 3 cm, the second with TOC between 200 and 500 DU, both variables following the distribution law as reported in Table 2 of Wandji Nyamsi et al. (2017). Figure 3a shows the influence of water vapor when deriving AOD from SDF. It shows that the spread of points slightly increases with higher burning threshold. Therefore, this demonstrates the importance of taking water vapor content into account. Figure 3b shows similarly the influence of ozone. The results clearly indicate that ozone does not have a significant influence, regardless of the threshold. Thus, a monthly climatology of ozone content is enough to infer AOD from SDF.

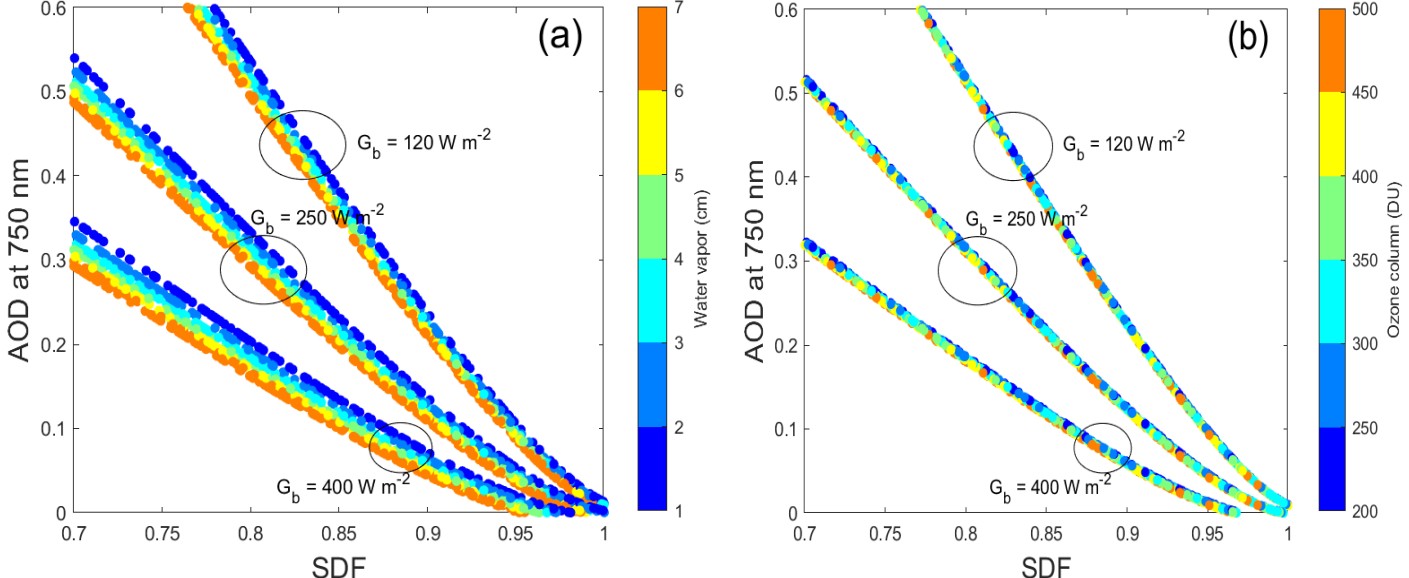

**Figure 3**: Same as Fig. 2 but varying (a) water vapor and (b) ozone contents.

## 5. Methodology

To ensure a reasonable set of ground-based stations as well as a good quality of measurements, four criteria have been applied on those ECA&D stations having both SD and TCC measurements within a calendar day. First, to include at least the brightening period, ECA&D stations having SD measurements starting before the year 1985 were chosen. Second, ECA&D stations should be collocated with AERONET stations with a maximum distance of 50 km. Third, the overlapping period that both cloud-free SD and AERONET measurements are available should cover at least three years. Finally, as the

fourth criteria, temporal homogeneity tests have been applied on SD measurements to select stations with homogeneous time series. For this last one, three main tests are used: the Standard normal homogeneity test (Alexandersson, 1986), the Buishand range test (Buishand, 1982) and the Pettitt test (Pettitt, 1979). If two of these three tests indicate that the time series is homogeneous with a confidence of 95%, then the station is included in the study. These filters result in a set of 16 stations over Europe. The station name, code, geographical coordinates and period from AERONET are indicated in Table 1. Table 2 reports similar details as in Table 1 but for SD and TCC measurements from the ECA&D database.

**Table 1**: Description of AERONET stations used and ordered by decreasing latitude

| ID # | Station name | Country | Lat. (°) | Long. (°) | Altitude (m) | Period |
|---|---|---|---|---|---|---|
| 1 | Munich University | Germany | 48.15 | 11.57 | 533 | 2002-01 to 2016-05 |
| 2 | Ispra | Italy | 45.80 | 8.63 | 235 | 1998-01 to 2010-04 |
| 3 | Montsec | Spain | 42.05 | 0.73 | 1574 | 2012-03 to 2016-06 |
| 4 | Valladolid | Spain | 41.66 | −4.71 | 705 | 2012-07 to 2016-05 |
| 5 | Zaragoza | Spain | 41.63 | −0.88 | 250 | 2013-01 to 2017-01 |
| 6 | Barcelona | Spain | 41.39 | 2.12 | 125 | 2004-12 to 2017-05 |
| 7 | Palma de Mallorca | Spain | 39.55 | 2.62 | 10 | 2011-08 to 2016-02 |
| 8 | Burjassot | Spain | 39.51 | −0.42 | 30 | 2007-04 to 2016-11 |
| 9 | Caceres | Spain | 39.48 | −6.34 | 397 | 2005-07 to 2012-05 |
| 10 | Badajoz | Spain | 38.88 | −7.01 | 186 | 2012-07 to 2016-06 |
| 11 | Murcia | Spain | 38.00 | −1.17 | 69 | 2012-10 to 2017-04 |
| 12 | Granada | Spain | 37.16 | −3.60 | 680 | 2004-12 to 2016-05 |
| 13 | El Arenosillo | Spain | 37.10 | −6.73 | 0 | 2000-02 to 2016-09 |
| 14 | Malaga | Spain | 36.71 | −4.48 | 40 | 2009-03 to 2016-07 |
| 15 | Santa Cruz Tenerife | Spain | 28.47 | −16.25 | 52 | 2005-07 to 2014-01 |
| 16 | Izana | Spain | 28.31 | −16.50 | 2391 | 2004-07 to 2015-09 |

The SD time series are divided into two periods, before and during the overlapping temporal coverage when both AOD and SD measurements are simultaneously available. The later period is called "learning/training period", while the earlier period with only SD measurements and no aerosol information is called "reconstruction period". A cloud-free day is assumed when the mean daily value of TCC is rounded to 0 okta. For each station of the Table 2, daily means are obtained from the average of at least 3 observations per day taken at around 06:00, 12:00 and 18:00 UT. Another way for cloud screening would be to use remote sensing cloud cover products. Unfortunately, this type of data suffers from a lack of temporal availability especially before the 1980s and their spatial resolution is neither good enough. This is why using ground-based measurements of total cloud cover is still the most reasonable way for the selection of cloud-free days for the scope of this study. However, in order to improve the selection of cloud-free days, an additional criteria was applied as follows: the presence of

cloud can be assumed if SDF is less than 0.7 (WMO, 2008). The uncertainty of SD measurement is 0.1 h (WMO, 2008) and the one of AOD at 750 nm is 0.01 (Eck et al., 1999). Because we assume that the effective wavelength in BAOD is close to AOD at 750 nm, any AOD value is converted to the corresponding AOD at 750 nm by means of the Ångström law. Hereafter, AOD at 750 nm is called AOD for the sake of simplicity.

**Table 2**: Description of ECA&D stations used providing SD/TCC measurements following the ID # of the station from Table 1. The distance between ECA&D and AERONET station is given.

| ID # | Station name | Lat. (°) | Long. (°) | Altitude (m) | Starting date Year–Month | Distance (km) |
|------|--------------|----------|-----------|--------------|--------------------------|---------------|
| 1 | Munchen-Bogenhausen | 48.14 | 11.60 | 521 | 1955–04 | 2.1 |
| 2 | Lugano | 46 | 8.97 | 273 | 1901–01 | 34.2 |
| 3 | Lleida | 41.63 | 0.59 | 192 | 1983–02 | 48.6 |
| 4 | Valladolid | 41.65 | –4.767 | 735 | 1973–10 | 5.3 |
| 5 | Zaragoza (Airport) | 41.66 | –1.01 | 247 | 1951–01 | 10.9 |
| 6 | Barcelona (Fabra Obs.) | 41.42 | 2.12 | 412 | 1974–01 | 3.6 |
| 7 | Palma de Mallorca (CMT) | 39.56 | 2.63 | 3 | 1972–08 | 0.3 |
| 8 | Valencia (Airport) | 39.49 | –0.47 | 69 | 1966–01 | 5.3 |
| 9 | Caceres | 39.48 | –6.367 | 459 | 1983–01 | 2.1 |
| 10 | Badajoz (Talavera la Real) | 38.88 | –6.83 | 185 | 1955–04 | 15.7 |
| 11 | Murcia | 38.00 | –1.17 | 61 | 1984–04 | 0.2 |
| 12 | Granada | 37.17 | –3.63 | 687 | 1941–06 | 3.8 |
| 13 | Huelva (Ronda del Este) | 37.28 | –6.91 | 19 | 1984–06 | 25.0 |
| 14 | Malaga (Airport) | 36.67 | –4.49 | 7 | 1947–12 | 5.4 |
| 15 | Santa Cruz de Tenerife | 28.46 | –16.25 | 35 | 1931–01 | 1.4 |
| 16 | Izana | 28.31 | –16.50 | 2371 | 1920–01 | 0.0 |

Within the learning period, the slope and intercept of the LRM method for each station are seasonally computed from the orthogonal-distance regression (ODR) fitting weighted by the measurement uncertainty ratio. There are four seasons where each season is grouped in three months as follows: December-January-February (DJF) for winter; March-April-May (MAM) for spring; June-July-August (JJA) for summer and September-October-November (SON) for autumn. Table 3 reports values for statistically significant relationships and used for the historical reconstruction of AOD. A linear relationship for a season is assumed statistically significant when the computed $p$-value between SDF and AOD is lower than 0.05.

Then, from the appropriate data sources as described in Section 2, daily atmospheric parameters are used to compute daily burning thresholds for the NHM method by using Eq. (2). For a given month, the burning threshold is computed as the median of at least seven daily values. Then, an

interpolation/extrapolation process to the nearest existing neighboring monthly threshold is used for the completion of data for cases of missing values. In such a case, an interpolated/extrapolated monthly value is kept if there is at least one monthly value within a season. After investigating several ways, it was found that the described process yields reasonable and physically understandable results. The computed local monthly burning threshold is then used as shown in Eq. (5) for the historical reconstruction of AOD.

**Table 3**: Number of cloud-free days (N), correlation coefficient (CC), slope (a), intercept (b) of the affine functions for each season between SDF and AOD following ID # station from Table 1. Only values for statistically significant relationships are reported.

| ID # | Winter | | | | Spring | | | | Summer | | | | Autumn | | | |
|---|---|---|---|---|---|---|---|---|---|---|---|---|---|---|---|---|
| | N | CC | a | b | N | CC | a | b | N | CC | a | b | N | CC | a | b |
| 1 | 27 | -0.59 | -0.40 | 0.42 | 74 | -0.59 | -2.11 | 2.08 | 59 | -0.77 | -1.16 | 1.16 | 33 | -0.66 | -0.90 | 0.93 |
| 2 | 145 | -0.23 | -4.56 | 3.73 | 54 | -0.61 | -2.93 | 2.69 | 37 | -0.39 | -3.48 | 3.01 | | | | |
| 3 | 37 | 0.34 | 0.04 | -0.02 | 42 | -0.39 | -3.72 | 3.54 | 59 | -0.65 | -2.41 | 2.30 | | | | |
| 4 | | | | | 17 | -0.61 | -0.35 | 0.35 | 63 | -0.40 | -1.62 | 1.52 | | | | |
| 5 | | | | | | | | | 56 | -0.63 | -2.03 | 1.94 | 18 | -0.74 | -1.37 | 1.33 |
| 6 | 49 | -0.53 | -1.62 | 1.63 | 38 | -0.34 | -1.99 | 1.86 | 34 | -0.73 | -4.18 | 3.88 | | | | |
| 7 | | | | | | | | | 71 | -0.66 | -5.21 | 4.74 | | | | |
| 8 | 62 | -0.50 | -1.54 | 1.50 | 46 | -0.77 | -1.11 | 1.12 | 103 | -0.73 | -1.57 | 1.55 | 27 | -0.59 | -0.76 | 0.78 |
| 9 | 83 | -0.27 | -0.16 | 0.17 | 68 | -0.31 | -1.41 | 1.36 | 143 | -0.54 | -2.29 | 2.22 | 74 | -0.36 | -1.70 | 1.57 |
| 10 | | | | | 60 | -0.30 | -2.32 | 2.23 | 174 | -0.49 | -3.53 | 3.40 | | | | |
| 11 | 48 | -0.42 | -0.40 | 0.40 | | | | | 80 | -0.66 | -3.86 | 3.72 | 30 | -0.68 | -1.03 | 1.01 |
| 12 | 154 | -0.47 | -0.36 | 0.36 | 95 | -0.35 | -1.85 | 1.75 | 329 | -0.65 | -2.57 | 2.48 | 165 | -0.50 | -1.32 | 1.24 |
| 13 | 243 | -0.34 | -0.49 | -0.49 | 260 | -0.38 | -0.61 | 0.62 | 397 | -0.49 | -2.79 | 2.62 | 254 | -0.28 | -1.70 | 1.56 |
| 14 | | | | | 60 | -0.56 | -1.80 | 1.70 | 200 | -0.70 | -2.97 | 2.77 | 21 | -0.69 | -2.07 | 1.93 |
| 15 | 63 | -0.31 | -2.98 | 2.62 | 48 | -0.35 | -4.95 | 4.37 | 85 | -0.92 | -3.63 | 3.42 | | | | |
| 16 | 241 | -0.32 | -0.12 | 0.13 | | | | | 277 | -0.61 | -3.38 | 3.33 | 258 | -0.22 | -1.17 | 1.13 |

All three methods are applied as described previously to show their individual performance to infer AOD from SD measurements. For each method, the deviations are computed by comparing to AERONET measurements. They are synthesized by the mean difference (MD), the root mean square difference (RMSD) and the correlation coefficient (CC).

$$MD = \frac{1}{n} \sum_{j=1}^{n} AOD_{method\ (j)} - AOD_{AERONET\ (j)} \tag{6}$$

$$RMSD = \sqrt{\frac{1}{n} \sum_{j=1}^{n} \left( AOD_{method\ (j)} - AOD_{AERONET\ (j)} \right)^2} \tag{7}$$

For each station, seasonal means were computed as the average of all available estimated daily AODs for cloud-free days over a given season. Then, seasonal anomalies were also computed as differences

between seasonal means and long-term mean obtained as the average of all seasonal means over the 1960-2010 period including the dimming/brightening period. The anomalies represent the quantity of interest in this study, as uncertainties and systematic biases associated with aerosol estimates can be then minimized, allowing more accurate interpretation of the changes in the past aerosol loads. The results of this part are analyzed and discussed in section 6.4.

## 6. Results and discussion

### 6.1. 6-hourly *vs* 1-hourly cloud cover data for the selection of cloud-free days

In this study, a cloud-free day is assumed when the mean daily value of TCC is rounded to 0 okta. This average value is computed from three-daily observations, typically at 06:00, 12:00 and 18:00 UT. Obviously, this is a very low temporal resolution for the selection of cloud-free days because within a 6-hour period, the sky can be partially or absolutely obscured by clouds such as cirrus, stratus or cumulus clouds resulting in an erroneous selection. Therefore, it is relevant to quantify how accurate the selection algorithm is when using 6-hourly observations instead of a much higher temporal resolution such as 1-hourly observations. In other words, which proportion of days with cloud influence on? SD is included in the set of clear sky days based on our selection algorithm?

In order to achieve this goal, we selected the well-known KNMI database because of the availability of metadata. This latter allows avoiding as much as possible inhomogeneity problems caused by station movements and changes in instruments. The KNMI database consists of hourly observations of numerous meteorological parameters from about 30 ground-based stations. The automatized hourly cloud cover measurements can be downloaded through the website http://projects.knmi.nl/klimatologie/uurgegevens/selectie.cgi. We used the temporal coverage from 2010 onwards. For the sake of clarity, let TCC1 and TCC6 denote the daily averages computed from 1-hourly and 6-hourly data, respectively. According to the WMO guidelines, TCC of 0 okta corresponds to a completely cloud-free sky, so it was considered that the limit of 1 okta could be extended to represent almost cloud-free sky conditions representing cases when clouds do not obstruct the SD measurements.

For each station, we retained days obeying to these following filters: (1) no missing hourly TCC between sunrise and sunset, (2) no missing TCC values at 06:00, 12:00 and 18:00 UT and (3) rounded TCC6 is equal to 0. Therefore, we investigated the set of remaining days. Figure 4 shows the distribution of frequency of days over the Netherlands. The x-axis is the okta bin with a width of 0.5 based on TCC1 indicated by the upper limit and the y-axis is the maximum value from the 1-hourly cloud observations between sunrise and sunset.

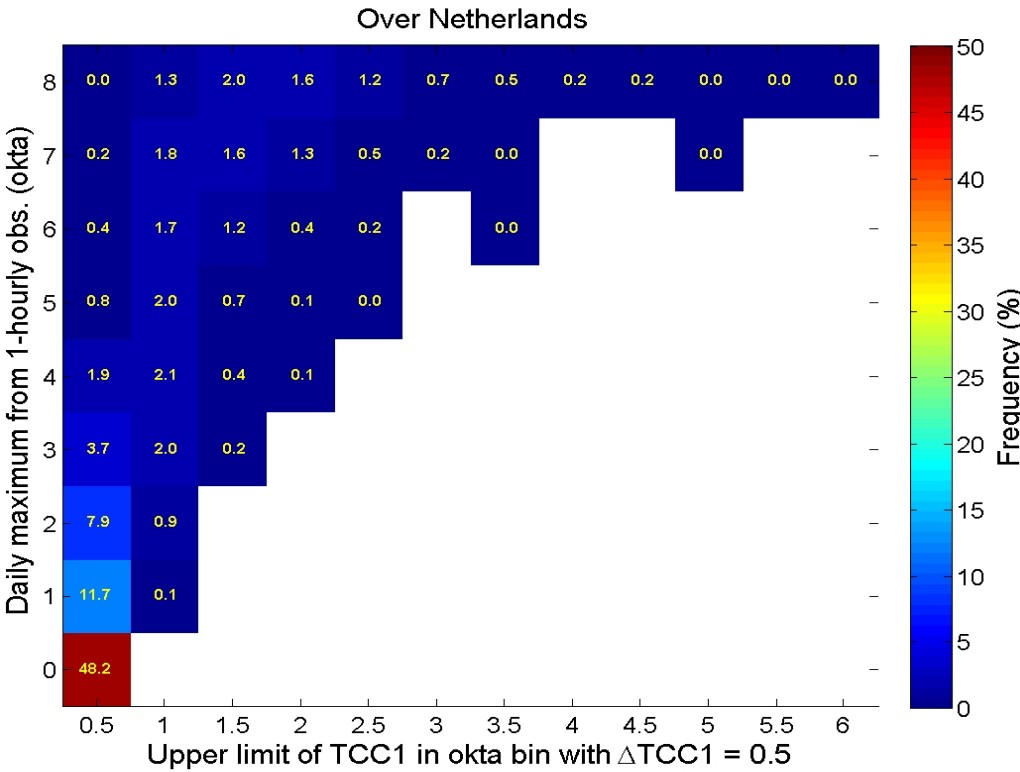

Figure 4. Relative frequency of days per okta bin with respect to the maximum cloud cover measurements between sunrise and sunset.

Regarding the TCC1 threshold of 1 okta for almost cloud-free days, one can observe that the selection algorithm successfully captures cloud-free days in 87% of the cases. Only 13% of the cases have a significant cloud influence. This demonstrates the good performance of the selection algorithm. For these 13% of the cases, the retrieved AOD are overestimated by cloud contamination whose impact is considered in the uncertainty estimates. For more detailed information on the uncertainty estimates, please refer to the Section 6.3.

### 6.2. Seasonality of burning threshold

Figure 5 shows the monthly burning thresholds for 5 selected stations: Malaga, El Arenosillo, Granada, Burjassot and Munich University. In general, there is a clear seasonality in the burning threshold, most clearly in Malaga, El Arenosillo and Granada. Depending on the station, the burning threshold ranges from high values (reaching up to 500 W m$^{-2}$ in January for Malaga) during the wintertime to low values (close to 120 W m$^{-2}$ for Burjassot and Munich) around the summertime. See Table S1 in the supplement for computed monthly burning threshold for all stations. The seasonality of the burning threshold exhibits an opposite seasonality as compared to temperature/humidity for the Northern hemisphere. In other words, the lower the temperature, the higher the burning threshold.

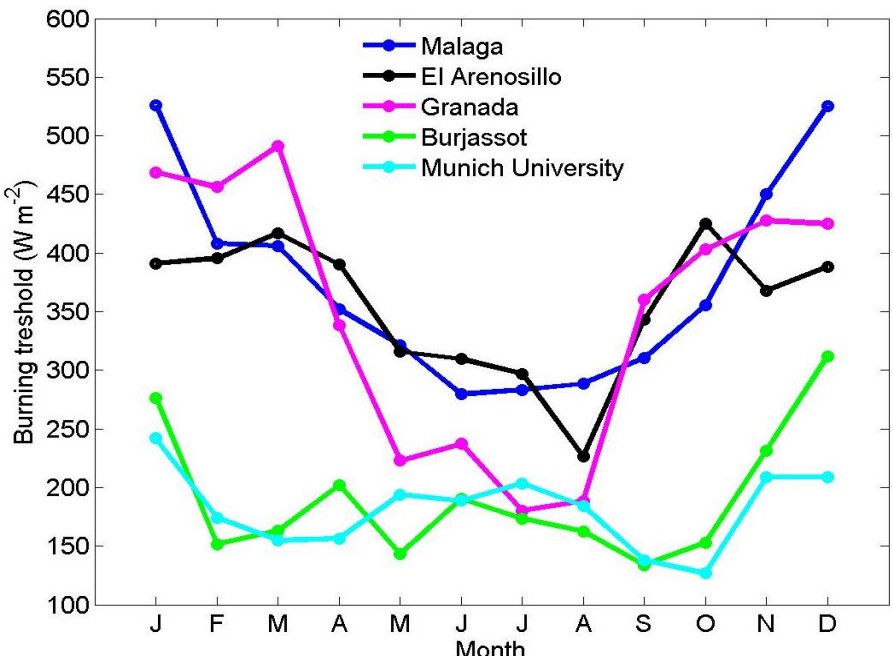

**Figure 5**. Monthly climatology of burning thresholds for 5 selected stations.

### 6.3. Performance of the three methods

To assess the performance of the methods, the available data of the most recent period are randomly divided into two datasets. One dataset is used to establish the slope and intercept for the LRM method. Then, the second dataset is used for the validation of all methods. The process is repeated several times to ensure the consistency of the validation results. Only stations with more than 80 data pairs for validation are retained to assess the performance of the three methods, resulting in the following 10

stations: Munich University, Ispra, Burjassot, Caceres, Badajoz, Murcia, Granada, El Arenosillo, Malaga and Izana.

Figure 6 shows an example of scatterplot between measured daily AOD from AERONET (horizontal axis) and estimated AOD from the three methods (vertical axis) over the validation period in Granada. This station is chosen because it well represents the features seen in most of the stations and seasons. The red, blue and green dots indicate the PBM, LRM and NHM methods, respectively and the dash colored line represents their corresponding linear fit.

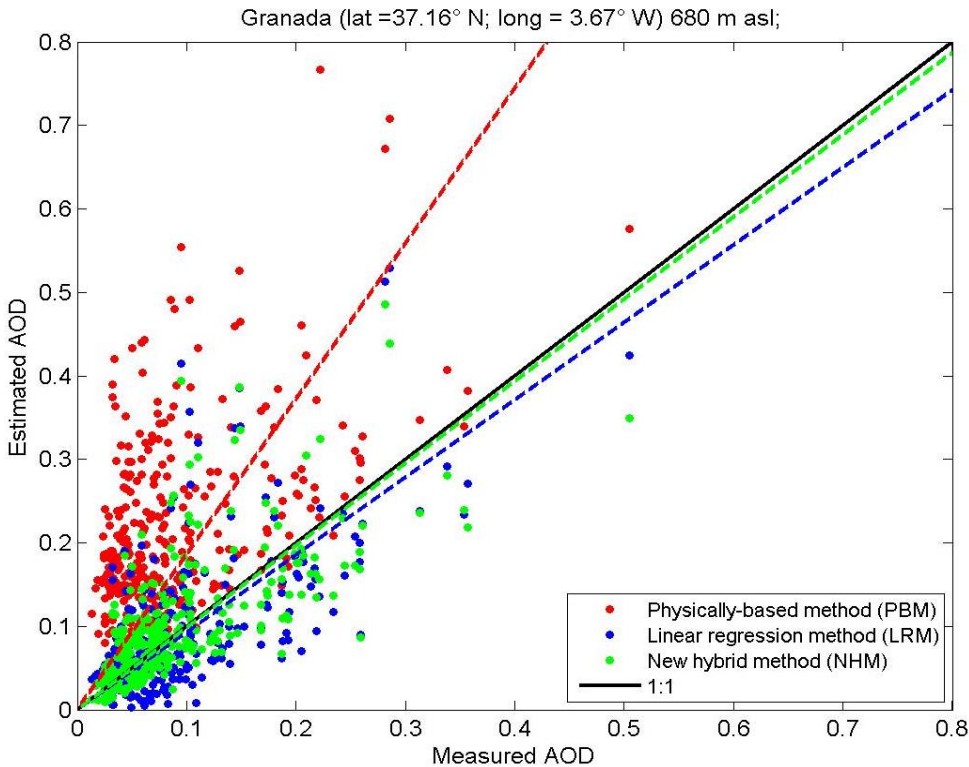

**Figure 6**: Scatterplot between measurements from AERONET and estimates from the PBM, LRM and NHM methods of daily AOD for Granada, Spain. The dash colored line represents their corresponding linear fit

The points from LRM and NHM in the graph follow quite well the 1:1 line with a limited spread. In contrast, there is a significant overestimation by PBM. The CC for PBM, LRM and NHM are 0.52, 0.66 and 0.73 respectively, thus indicating that the variability in AOD is better captured by NHM and LRM than by PBM. The biases for NHM and LRM are very close to zero while PBM shows a noticeable positive bias of 0.14. This confirms that the use of constant 120 W m$^{-2}$ as a burning threshold is not

necessarily suitable when deriving AOD from SD measurements. The RMSD for NHM and LRM are quite similar and small (approximately 0.06) and differ noticeably from the one of PBM of 0.17.

Depending on the statistical indicators, the best performance is seen either with the LRM method or with the NHM method. At all stations, the worst performance is seen with the PBM method, largely due to the errors induced by the use of 120 W m$^{-2}$ as a burning threshold.

**Table 4**: Statistical indicators for each station. N is the number of data pairs. Mean is the average of AERONET measurements. MD is the mean difference. RMSD is the root mean square difference. The first value is the PBM method, the second one is the LRM method and the third one is the NHM method. The best performance is in bold.

| Station | N | Mean | CC | MD | RMSD |
|---|---|---|---|---|---|
| Munich | 88 | 0.07 | 0.60 / 0.60 / **0.63** | 0.04 / **0.00** / **0.00** | 0.07 / **0.04** / **0.04** |
| Ispra | 98 | 0.10 | 0.42 / 0.28 / **0.65** | 0.19 / 0.07 / **–0.02** | 0.21 / 0.17 / **0.07** |
| Burjassot | 110 | 0.09 | 0.60 / 0.62 / **0.65** | 0.07 / **0.00** / 0.02 | 0.13 / **0.07** / 0.09 |
| Caceres | 120 | 0.06 | 0.36 / 0.40 / **0.55** | 0.10 / 0.00 / **–0.00** | 0.13 / 0.04 / **0.04** |
| Badajoz | 97 | 0.08 | 0.40 / 0.35 / **0.43** | 0.07 / 0.02 / **0.00** | 0.10 / 0.07 / **0.06** |
| Murcia | 82 | 0.09 | 0.58 / 0.60 / **0.63** | 0.06 / 0.01 / **–0.00** | 0.11 / 0.12 / **0.08** |
| Granada | 344 | 0.09 | 0.52 / 0.66 / **0.73** | 0.14 / **0.00** / 0.01 | 0.17 / **0.06** / **0.06** |
| El Arenosillo | 584 | 0.09 | 0.34 / 0.32 / **0.37** | 0.21 / **0.01** / 0.02 | 0.28 / **0.10** / 0.11 |
| Malaga | 137 | 0.12 | 0.68 / 0.66 / **0.70** | 0.15 / **0.00** / **0.00** | 0.17 / **0.07** / **0.07** |
| Izana | 349 | 0.08 | 0.28 / **0.63** / 0.60 | 0.17 / **0.00** / 0.03 | 0.31 / **0.08** / 0.13 |
| **Averaged statistics** | | | 0.48 / 0.51 / **0.59** | 0.12 / 0.01 / **0.00** | 0.17 / 0.08 / **0.07** |

Table 4 reports the statistical indicators summarizing the errors of the three methods for each station. In general, the performance of LRM and NHM is quite similar and clearly better than the PBM. For all methods, the CC varies between 0.3 and 0.7, with best correlation observed mostly with the NHM method. The MD for PBM is 0.04 (in Munich) at the minimum and reaches up to 0.2 (in El Arenosillo). In all stations, both LRM and NHM show biases mostly close to zero in terms of MD. The RMSD reaches up to 0.3 (in Izana as well as El Arenosillo) for PBM, while for both LRM and NHM, it reaches up to 0.1 denoting a more limited spread of points. Overall, the NHM performs slightly better than the LRM as indicated by the averaged statistics. Since PBM shows the worst performance in all stations, in the following, we exclude this method in all further analysis.

A diagnostic method is used to quantify the uncertainties corresponding to sunshine duration based AOD retrievals (Sayer et al., 2020). AERONET observations are used to derive the diagnostic expected error ($EE_{AOD}$) envelope for the retrieved AOD. This type of EE envelope uncertainty estimate is similar to the corresponding one of MODIS Dark Target satellite AOD retrievals (Levy et al., 2010). As derived using

AERONET AOD as ground-truth, the diagnostic EE envelope includes all possible sources of uncertainties due to for example, changes of the card type, burning threshold, cloud contamination, changes in aerosol properties during the day, and uncertainties in sunshine duration measurements. To derive the EE estimates, we use a random subset of about 7000 AOD retrievals from the validation dataset as described previously with all stations. We divide the data into 100 bins that each bin contains same amount of measurements. We compute the standard deviation of the retrieval error and the average of retrieved AOD in each bin, see Figure 7. We intentionally select the retrieved AOD as the variable to compare the uncertainty against so that it is possible to estimate the retrieval uncertainties also in cases in which accurate or measured AOD is not available. We fit a linear model to the uncertainty data and derive the EE envelope estimate as follows:

$$EE_{AOD} = \pm (0.01 + 0.40 \times AOD_{NHM}) \tag{8}$$

where $AOD_{NHM}$ is the retrieved AOD from the NHM method.

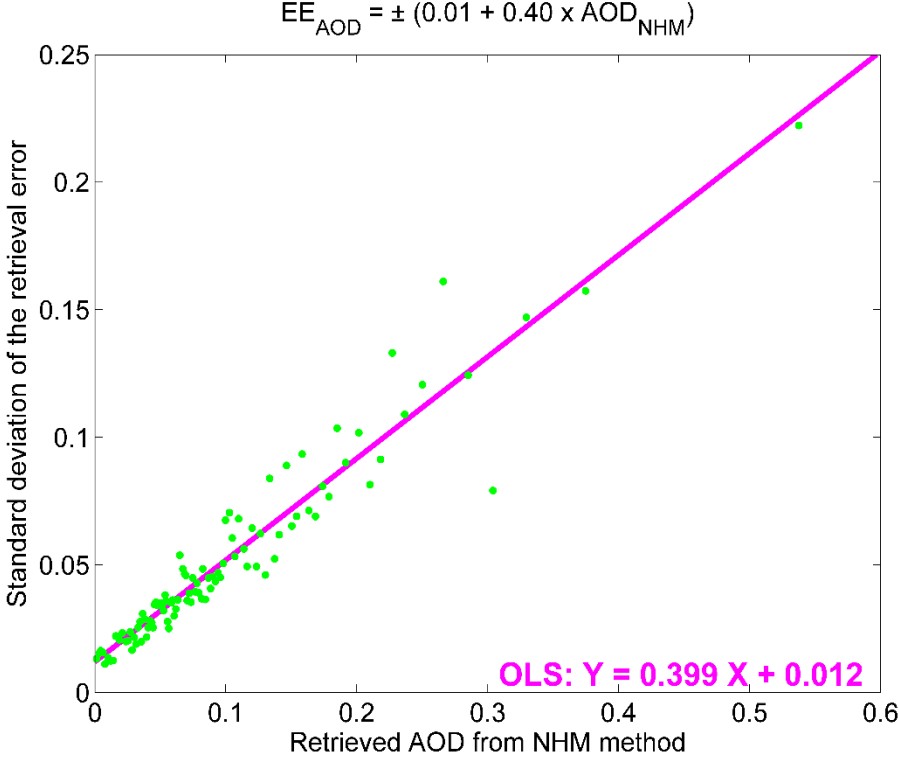

**Figure 7**: Scatter plot showing the standard deviations of the retrieval error as function of the average of retrieved AOD in each bin. The magenta line represents the fitted uncertainty estimate

## 6.4. Reconstructed historical evolution of aerosol load

The slopes and intercepts of the LRM method in Table 3 are used to derive AOD from SDF for all cloud-free days over the period where SD and TCC are commonly available as reported in Table 2. Figure 8 shows an example of the historical evolution of seasonal mean AOD in Malaga from five data sources. The pink, blue, green, orange and brown lines indicate the seasonal mean AOD time series from AERONET, LRM, NHM, MERRA-2 and MODIS respectively.

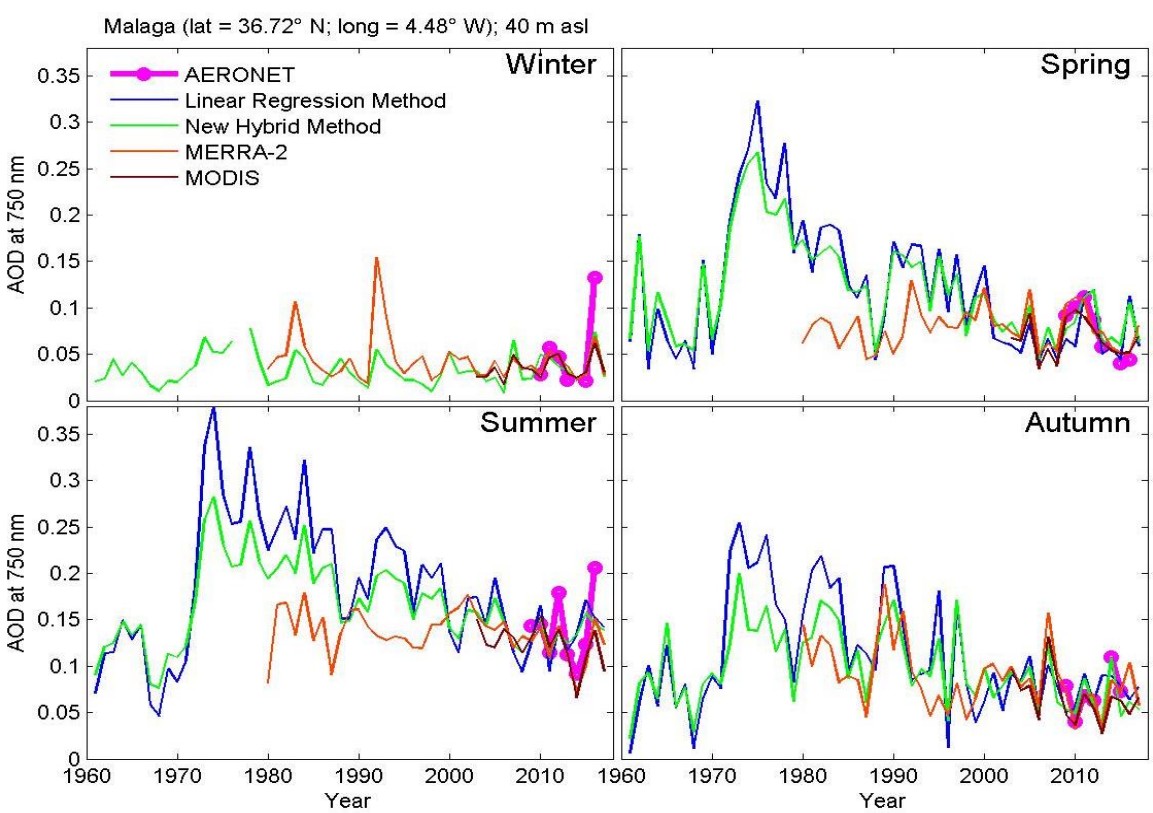

**Figure 8**: Time series of the seasonal mean AOD from five different data sources in Malaga, Spain.

There is no AOD estimate from the LRM method in wintertime because the CC of the linear fit was not statistically significant. However, the NHM method is able to provide AOD estimates, clearly showing the potential of the NHM method to operate in all seasons. In general, the estimated AOD values from the different sources are relatively close to each other depending on the season and the period. When compared to AERONET AOD available from 2009 onwards, all AOD estimates follow quite closely the measurements during all seasons. Over the full period, the LRM curve follows rather closely the NHM curve in springtime while less so in summer and autumn but nevertheless exhibits a quite similar inter–annual variability and overall trends. The 2016 peak values observed in wintertime and summertime from

AERONET measurements are due to the fact that only February and mostly June measurements, respectively, are used to compute the seasonal means. Over those months, few high AODs were measured. Some deviations are seen between the AOD estimates such as those from methods. This is the case, for instance, between the LRM and NHM methods in summer. Therefore, further investigations were carried out in more details.

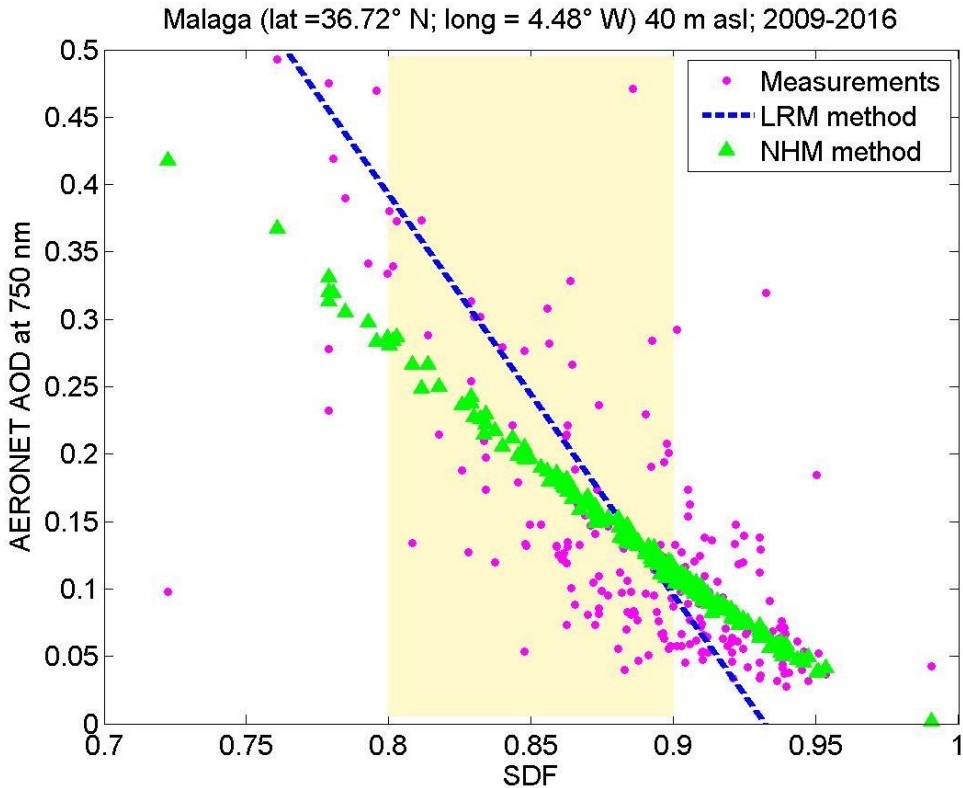

**Figure 9**: Scatterplot between SDF and AOD for Malaga, Spain. Regression line from LRM is in dash line. NHM estimates are in green triangle.

Between the mid-1970s and the 2000s, the AOD estimates from LRM tend to be greater than those from NHM in Malaga. This disparity can be explained as follows: when applying the LRM method, a fixed AOD value derived from a given SDF can be lower or greater than the true AOD value. Figure 9 shows the cloud of magenta dots corresponding to ground-based measurements between SDF and AOD in Malaga during the summer season. The linear fit obtained from the LRM method is in blue dashed line and the estimate from NHM is in green triangle. The period between the mid-1970s and the 2000s in summertime of Fig. 8 is typically characterized by SDF ranging between 0.8 and 0.9 delimited by the yellow area in Fig. 9. A visual inspection shows that the vast majority of the dots within the yellow area is below the regression line meaning that the measured AOD values are lower than the estimated AOD

values derived from the LRM method (blue dashed line). In other words, any estimates lower than those from LRM method could be considered as more realistic, which is the case with the NHM estimates. In Fig. 8, we can observe that satellite-derived estimates especially in MERRA-2 over the period 1980-2000 differ noticeably from the LRM and NHM estimates. Prior to year 1999, however, MERRA-2 assimilated

AOD only from AVHRR measurements and only over ocean. Randles et al. (2017) reported that the standard deviation of AOD in summer in MERRA-2 does not compare well to the observations at the Goddard Space Flight Center (GSFC), Maryland in USA due to the anthropogenic emissions of the model. The Malaga station is located in an urban area so it may be possible that the MERRA-2 at Malaga suffers from similar problems as reported in GSFC explaining the difference between MERRA-2 and

SD-based AOD.

It is worth mentioning that, in addition to ODR, we have tested several other regression methods such as ordinary least squares (OLS). We found that depending on the season, the reconstructed aerosol load from the OLS method can agree very well with the one from NHM resulting in a close to zero average difference over the full period. However, this was not generally true and in some cases there were

significant differences between different regression methods. This highlights the importance to select the most appropriate method that takes into account the measurement uncertainties (Mikkonen et al., 2019).

One of the longest time series for both SD and TCC measurements is from Izana. The results for this station are shown because there are several studies in the literature providing the historical evolution of summer season AOD derived with different approaches, thus allowing now further comparisons and

discussions. Figure 10 shows summer season means of AOD estimated from five data sources from 1955 onwards since a breakpoint was found in 1953 within the SD measurements.

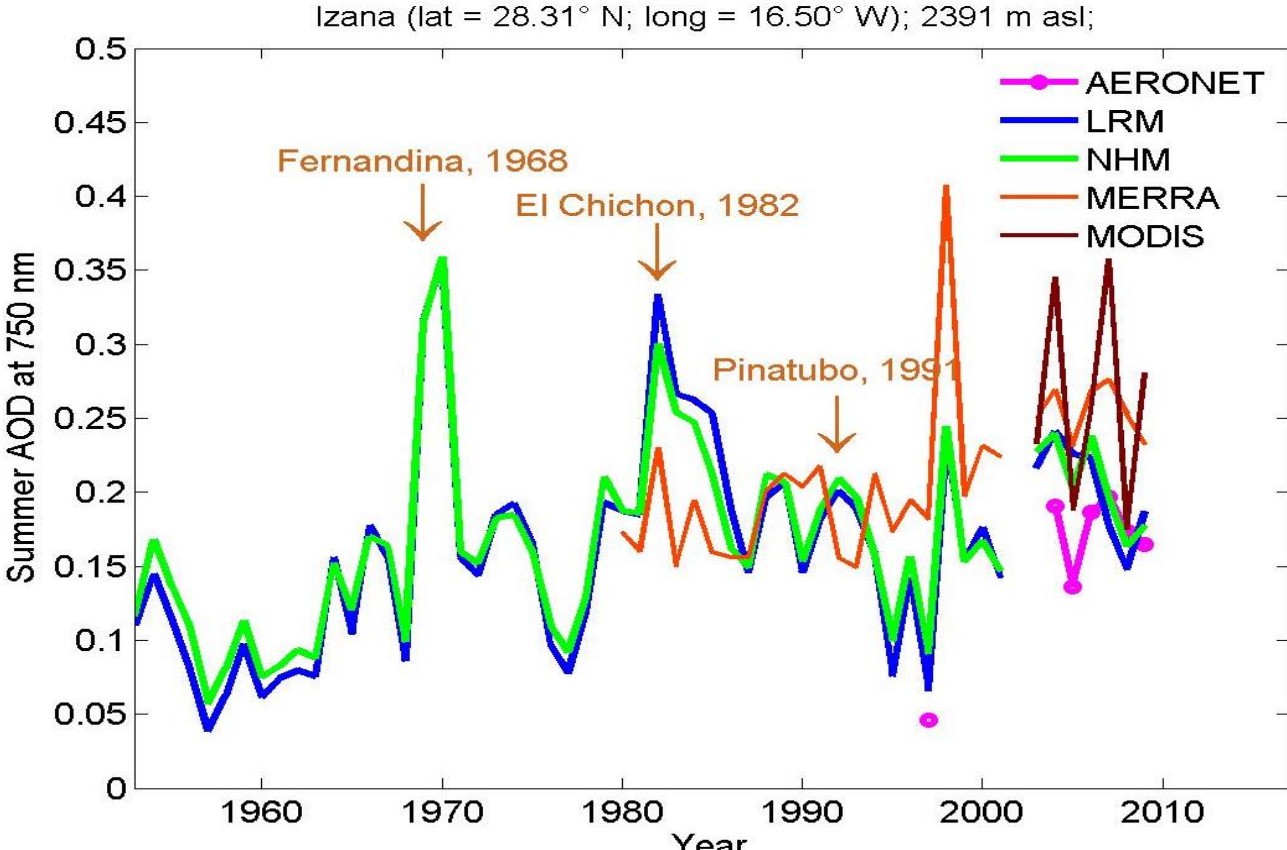

**Figure 10**: Time series of the summer means of AOD in Izana, Spain.

In the Figure 10, there are no AOD estimates in the year 2002 due to missing SD measurements from November 2001 to December 2002 at Izana. Satellite observations are not either displayed for the year 2002. Overall, both MERRA-2 and MODIS, exhibit an overestimation at Izana as compared to reconstructed AOD by means of the LRM and NHM methods. In site-by-site comparisons, there was

5      also a clear overestimation when MODIS and MERRA-2 estimates were compared against AERONET ground-based measurements (not shown).

Both LRM and NHM AOD show a noticeable and quite similar temporal variability and extreme values at the Izana site. Both methods are sensitive to known episodic aerosol events, the peak values observed in AOD estimates from NHM are well in agreement with important volcanic eruptions such as Arenal

10     and Fernandina Island in 1968, Chichón in 1982 and Pinatubo in 1991. We compared our results for Izana obtained with the NHM method during summer with AOD at 500 nm obtained by Garcia et al. (2016) from an Artificial Neural Network using input parameters such as visibility, SD, TCC, relative

humidity and temperature. Both approaches agree well with a minimum around 1955. Then, the average AOD increases until around 1982 and followed by a decreasing trend until around 2000 (not shown).

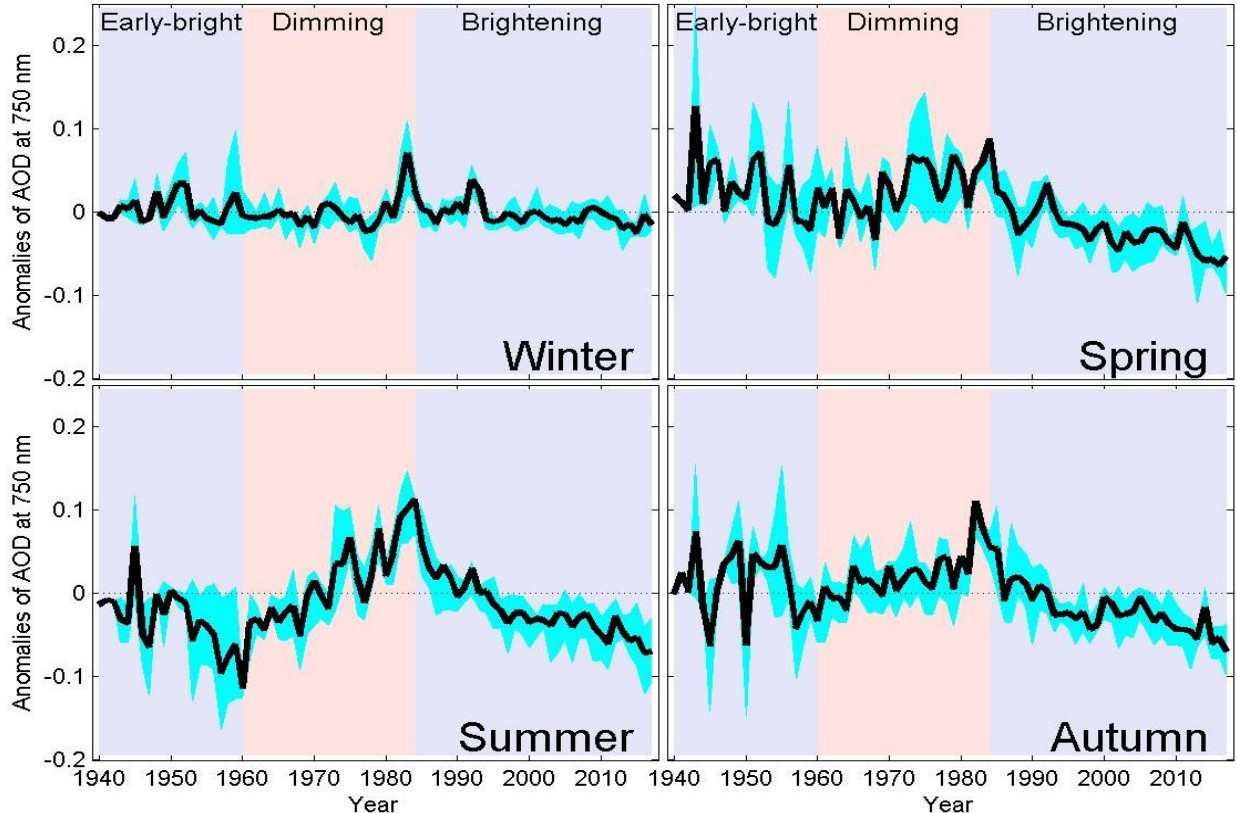

**Figure 11**: Evolution of the mean of seasonal anomalies of AOD for all stations given in Table 1. The cyan area refers to range between 1st and 3rd quartiles of the different series for each year. The light purple and misty rose bands indicate (early) brightening and dimming periods, respectively.

Figure 11 shows the mean of seasonal anomalies (black line) of AOD for all stations given in Table 1. The lower and upper shading limits represent 1st and 3rd quartiles respectively. The peak values are mostly around the specific years when aerosol events are known to be widespread at global scale.

During the winter season, there are no clear periods of increasing and decreasing trends. However, in all the other seasons, both dimming and brightening periods can be seen and can be summarized by three main phases as follows:

- From the 1940s to the late 1950s, the anomalies show a smooth decrease with the most pronounced decrease during the summer season. This behavior is known as early brightening

with a peak around the late 1940s and agrees with earlier studies (Wild, 2009; Sanchez-Lorenzo et al., 2015).

- From the late 1950s to the late 1980s, a moderate increase of the anomalies is seen, known as dimming period. It is in agreement with the well-known period of reduction in SSI at global scale, widely stated in the literature (Stanhill and Cohen, 2001; Wild et al., 2005; Stjern et al., 2009; Wild, 2009).

- From the 1990s onwards, a decrease of the relative anomalies is seen. This is in agreement with previous findings of increasing SSI during this period, known as brightening period (Wild et al., 2005, 2008; Wild, 2009).

## 7. Conclusions

In this study, we have proposed a new method for estimating AOD from sunshine duration measurements. It is a physically-based method similar to the approach used with the sunphotometer measurements of AOD. The method is used for reconstructing the historical AOD under cloud-free conditions as far back to the past as possible using in addition to the sunshine duration measurements also daily total ozone column and total water vapor from ECMWF twentieth century reanalysis ERA-20C, which are available at global scale from 1900 to 2010. Surface synoptic cloud observations are further required to identify cloud-free days. In addition, as an input, it uses the seasonal burning thresholds, which are measurement station dependent.

We applied the method to sixteen ground-based stations over Europe. As a result, the reconstructed AOD time series shows a comprehensive seasonal variability. It improves the detection of the signal induced by aerosol events such as volcanic eruptions and the gradual changes caused for instance by air pollution compared to two earlier developed and published state-of-the-art methods in the literature. In addition, the time series are consistent with earlier published results, both with the dimming characterized by an increase in AOD over the 1960–1984 period and the subsequent brightening period with a decrease in AOD until the 2010s. An early brightening is partially observed from the 1940s until the late 1950s confirming some earlier findings based on limited studies. Taking into account the universality of the proposed method, this study opens the way to extend the reconstruction of the historical evolution of aerosol load before the mid-20th century to other regions of the world where both sunshine duration measurements and cloud information are available.

## A. Appendix: Equations for broadband direct irradiance model

$$G_b = \varepsilon\, G_o\, T_R\, T_g\, T_o\, T_w\, T_a \tag{A.1}$$

$$\varepsilon = 1 + 0.03344\, cos\left(\frac{2\,n}{365.2422} - 0.049\right) \tag{A.2}$$

$$T_R T_g = \left(1 - \frac{0.606\, m_R'}{6.43 + m_R'}\right)(1 - 0.0075\, m_R'^{\,0.875}) \tag{A.3}$$

$\quad$ $$T_o = exp\left(-0.0365\,(m_o l_o)^{0.7136}\right) \tag{A.4}$$

$$T_w = 1.0121 - 0.11\,(0.8\, m_w l_w + 0.00063)^{0.3} \tag{A.5}$$

$$T_a = exp\left(-m_a\beta(0.6777 + 0.1464\, m_a\beta - 0.00626\,(m_a\beta)^2)^{-1.3}\right) \tag{A.6}$$

$$m_R' = m_R\, exp\left(-\frac{z_o}{8430}\right) \tag{A.7}$$

$$m_R = \left(cos\,(\theta_s) + 0.45665\,\theta_s^{\,0.07}(96.4836 - \theta_s)^{-1.6970}\right)^{-1} \tag{A.8}$$

$\quad$ $$m_w = \left(cos\,(\theta_s) + 0.031141\,\theta_s^{\,0.1}(92.4710 - \theta_s)^{-1.3814}\right)^{-1} \tag{A.9}$$

$$m_o = \left(cos\,(\theta_s) + 268.45\,\theta_s^{\,0.5}(115.420 - \theta_s)^{-3.2922}\right)^{-1} \tag{A.10}$$

$$m_a = m_w \tag{A.11}$$

$n$ is the number of the day over the year starting from #1 for 1st January

$l_o$ is the total column content of ozone (DU)

$\quad$ $l_w$ is the total column content of water vapor (cm)

$\beta$ is the aerosol optical depth at 1000 nm

$z_o$ is the altitude of site above mean sea level (m)

**Data availability.** AERONET data used here are available from https://aeronet.gsfc.nasa.gov/new_web/download_all_v3_aod.html. Sunshine duration and total cloud cover measurements are available through the European Climate Assessment & Dataset project (https://www.ecad.eu). Products from ERA-20C ECMWF can be downloaded from the following website: https://www.ecmwf.int/en/forecasts/datasets/reanalysis-datasets/era-20c. Daily OMI TOCs are available from the website https://acdisc.gesdisc.eosdis.nasa.gov/data/Aura_OMI_Level3/OMTO3d.003/. MODIS MxD08_D3 products can be downloaded from the web site https://ladsweb.modaps.eosdis.nasa.gov/. Modern-Era Retrospective analysis for Research and Applications, Version 2 (MERRA-2) are available through the Goddard Earth Sciences Data and Information Services Center (GES DISC; http://disc.sci.gsfc.nasa.gov/mdisc/)

**Author contribution**. WWN designed and developed the presented method with help from all co-authors. WWN, AL and AA implemented the method. WWN, AL, ASL, MW and AA participated in writing and editing the manuscript, as well as investigating and interpreting the results.

**Supplement**. The supplement related to this article is available online at: xxxxxxxxx.

**Competing interests.** The authors declare that they have no conflict of interest.

**Acknowledgments.** The authors acknowledge the data providers within and the team leading the ECA&D project (https://www.ecad.eu). They thank the principal investigators and the co-investigators and their staff for establishing and maintaining the AERONET sites used in this investigation. They also thank Mikko R. A. Pitkänen and Antti Kukkurainen for their useful feedbacks on this study and Dr. Else van den Besselaar from Koninklijk Nederlands Meteorologisch Instituut (KNMI) for her help in the use of ECA&D data. The research leading to these results has received funding from the Academy of Finland under grant agreement no. 309497. A. Sanchez-Lorenzo was supported by a fellowship RYC-2016–20784 funded by the Spanish Ministry of Science and Innovation.

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
