# Peer review of "A hybrid method for reconstructing the historical evolution of aerosol optical depth from sunshine duration measurements"

_Atmospheric Measurement Techniques, 2019_

## Referee Comment (RC1) · Anonymous Referee #1 · 18 Jan 2020

Review of "A hybrid method for reconstructing the historical evolution of aerosol optical depth from sunshine duration measurements"

Information of long term series of atmospheric data is an essential task in order to understand various processes going back to more than 20-30 years. Aerosol optical depth investigated here, is a parameter that for surface based measurements is not available before the early 1990's. So the motivation of the paper to use SD mainly and other data in order to retrieve AOD in the past is solid and the results valuable for the atmospheric community.

As the authors quote, this is not the first paper approaching this issue (AOD retrieval)

[Figure]

using SD data, however a new and other methods are compared here.

My point of view is that there are two major aspects that are not mentioned in this work and without addressing them the work limits its credibility.

The first one is the uncertainty estimation. This is a difficult task but in order for the AOD data that will be calculated from past SD data to be useful, authors have to provide an uncertainty estimation. There are a lot of aspects linked with this uncertainty that have to do with:

- The input data.

There are various technical aspects mentioned and referring the Sanchez – Lorenzo work but the effect is not quantified. In addition to them, the use of daily information increases the uncertainty related with the presence of clouds. Datasets that were used for cloud cover assessment include various uncertainties, the simplest one being that their temporal resolution within one day can not ensure for the 100%-non presence of clouds.

- With the method itself

This is addressed but not quantified through the aeronet based comparison. Temporal issues from the use of daily data too.

The second one already mentioned partly, is the role of clouds.

a. Cirrus clouds. Cirrus clouds optical thickness (or depth) is in the same magnitude as the one of aerosols. In a number of cases they can be practically invisible and their effect on the SD measurements can not be distinguished with the one of the aerosols. In the case that instruments like sky cameras or human observations are used, the problem can be partly solved. However, data series going back to the past include mostly or only observations, commonly with a temporal resolution of the order of hours. Moreover, for meteorological station observers it is evident that SD measurements are not affected by partly visible or invisible cirrus clouds.

b. Clouds. During the course of the day the presence of clouds will have an effect on the SD duration data. Daily mean cloud cover used here from ECA & D is defined "Daily mean cloud cover: Whenever synoptical cloud cover data is available at 00, 06, 12 and/or 18 UT, mean daily cloud cover is calculated as the average of the available values." So CCT ∼0 is only an average of 4 measurements during the day. In addition, WMO has defined the SDF>0.7 for a cloudless condition threashold but of course for Aerosols where in most of the cases, the solar attenuation is well within those limits, the threashold serves eliminating only part of the number of cases that can introduce a cloud related uncertainty to the method application.

Moreover, AERONET related comparisons does not include most of cirrus related and all of thicker clouds cases as these are algorithmically eliminated from aeronet retrievals.

Summarizing, the presence of clouds in some part of the day (with SDF>0.7) can be comparable to the aerosol effect. This causes a small but systematic overestimation of the effect of aerosols in DNI measurements when applying any method based on radiative transfer modeling (where clouds are set to zero) to past series. In addition, periods in the past with changes in cirrus clouds (or increased partly cloudiness) could be wrongfully characterized based in this method, as high aerosol periods.

Minor comments

Page 2 L15 AERONET has the denser network but looking at the site I can see only four stations with data series more than 15 years. Probably WMO-GAW network or Skynet network can be mentioned too.

15 line 15 Ångström

Why is the effective wavelength in BAOD close to AOD at 750 nm ?

How can anyone trust Merra data for small to moderate aerosol changes in the past when Pinatubo and El Chichon are not visible in the dataset ?

---

## Referee Comment (RC2) · Anonymous Referee #2 · 24 Feb 2020

In this work, a new method for estimating AOD from SD measurements under cloud-free days is proposed. The adopted method relies on a broadband DNI model that takes into account the local conditions affecting SD measurements. The work is well suited to the readership of AMT. However, there are a number of issues with the procedure developed in the paper that prevent its publication in the current form. The main issues with the work concern the decision to use daily mean of cloud cover data obtained only from four observations terms (every 6-hours).

There are also other meteorological databases which contain a variety of hourly meteorological data from weather stations located in Europe (e.g.

[Figure]

https://www.ncdc.noaa.gov/isd). Therefore, the authors must quantify if there are significant differences between selecting the cloud-free days using hourly cloud data and ECAD dataset. Also, satellite cloud cover product data could be used in the analysis, as they provide hourly cloud cover data for more than twenty years (https://www.cmsaf.eu/EN/Products/AvailableProducts/Climate_Data_records/Climate_Data_Records_node.html)

Minor issues:

Page5Line20. The authors must mention the studies in which was performed the comparison between ECMWF reanalysis product and AERONET data.

Page6Line5. Which are the studies that demonstrate the performances of TOC estimates?

Page6Line15. The authors must motivate why they have used the MxD08_D3m Modis product instead of the higher spatial and temporal resolution Modis aerosols data (MxD04_L2 product).

Page10Line20. For which station the 12 000 the clear sky atmospheric states were built?

Page15Line5. The authors must discuss why most of the stations selected after applying the homogenization test are located in the Mediterranean climate. It is hard to believe that, excepting data from two stations, only the stations located in Spain provide homogenous SDF data.

Page18Line10. The authors must provide the burning thresholds computed for all the stations as an ESM table. This will add value to the paper.

Page19Line5. In order to better emphasise the results of the three models tested, trendlines for each model should be added on the scatterplot.

Some references mentioned in the text are missing from the Bibliography section (e.g. Qui, 1998).

---

## Author Comment (AC1) · 31 Mar 2020

ANSWERS TO REFEREE #1

First of all, we thank Referee #1 for these constructive remarks and comments. The comments have been addressed below and have been taken into account for revising a part of the text following recommendations of the referee. The responses to the reviewer points are below after the reviewer points that are in italics.

*Comment. Information of long term series of atmospheric data is an essential task in order to understand various processes going back to more than 20-30 years. Aerosol optical depth investigated here, is a parameter that for surface based measurements is not available before the early 1990's. So the motivation of the paper to use SD mainly and other data in order to retrieve AOD in the past is solid and the results valuable for the atmospheric community. As the authors quote, this is not the first paper approaching this issue (AOD retrieval) using SD data, however a new and other methods are compared here.*

*My point of view is that there are two major aspects that are not mentioned in this work and without addressing them the work limits its credibility.*

***The first one is the uncertainty estimation.** This is a difficult task but in order for the AOD data that will be calculated from past SD data to be useful, authors have to provide an uncertainty estimation. There are a lot of aspects linked with this uncertainty that have to do with:*

*-The input data.*

*There are various technical aspects mentioned and referring the Sanchez–Lorenzo work but the effect is not quantified. In addition to them, the use of daily information increases the uncertainty related with the presence of clouds. Datasets that were used for cloud cover assessment include various uncertainties, the simplest one being that their temporal resolution within one day cannot ensure for the 100%-non presence of clouds.*

*-With the method itself*

*This is addressed but not quantified through the aeronet based comparison. Temporal issues from the use of daily data too.*

**The second one already mentioned partly, is the role of clouds.**

a. *Cirrus clouds. Cirrus clouds optical thickness (or depth) is in the same magnitude as the one of aerosols. In a number of cases they can be practically invisible and their effect on the SD measurements cannot be distinguished with the one of the aerosols. In the case that instruments like sky cameras or human observations are used, the problem can be partly solved. However, data series going back to the past include mostly or only observations, commonly with a temporal resolution of the order of hours. Moreover, for meteorological station observers it is evident that SD measurements are not affected by partly visible or invisible cirrus clouds.*

b. *Clouds. During the course of the day the presence of clouds will have an effect on the SD duration data. Daily mean cloud cover used here from ECA&D is defined "Daily mean cloud cover: Whenever synoptical cloud cover data is available at 00, 06, 12 and/or 18 UT, mean daily cloud cover is calculated as the average of the available values." So CCT ~0 is only an average of 4 measurements during the day. In addition, WMO has defined the SDF>0.7 for a cloudless condition threshold but of course for aerosols where in most of the cases, the solar attenuation is well within those limits, the threshold serves eliminating only part of the number of cases that can introduce a cloud related uncertainty to the method application.*

*Moreover, AERONET related comparisons does not include most of cirrus related and all of thicker clouds cases as these are algorithmically eliminated from aeronet retrievals.*

*Summarizing, the presence of clouds in some part of the day (with SDF>0.7) can be comparable to the aerosol effect. This causes a small but systematic overestimation of the effect of aerosols in DNI measurements when applying any method based on radiative transfer modeling (where clouds are set to zero) to past series. In addition, periods in the past with changes in cirrus clouds (or increased partly cloudiness) could be wrongfully characterized based in this method, as high aerosol periods.*

Thank you for these two main remarks. First about the uncertainty estimation. The key science question, to be answered with our proposed method, is the following: what is the long-term historical evolution of the atmospheric aerosol load? Although the method can provide absolute daily aerosol estimates, the most relevant quantities therefore are the seasonal means and more especially their corresponding anomalies. This latter is widely used because uncertainties and systematic biases can be then minimized, allowing more accurate interpretation of the changes in the past aerosol loads. To make this point clearer also in the manuscript, we have included the following statement in the last paragraph of the "Methodology" section as follows:

"The anomalies represent the quantity of interest in this study, as uncertainties and systematic biases associated with aerosol estimates can be then minimized, allowing more accurate interpretation of the changes in the past aerosol loads."

Nevertheless, in order to quantify uncertainties associated with input data, the method itself and cloud information, diagnostic and prognostic (predictive) methods can be used. Here, we use the diagnostic method for the uncertainty estimates by comparing the sunshine duration retrievals to AERONET data (Sayer et al., 2020). In the revised manuscript, we have carried out the uncertainty estimates, and we have re-written the relevant part of the text after the last paragraph of Section 6.2 (Performance of the three methods) as follows:

"A diagnostic method is used to quantify the uncertainties corresponding to sunshine duration based AOD retrievals (Sayer et al., 2020). AERONET observations are used to derive the diagnostic expected error ($EE_{AOD}$) envelope for the retrieved AOD. This type of EE envelope uncertainty estimate is similar to the corresponding one of MODIS Dark Target satellite AOD retrievals (Levy et al., 2010). As derived using AERONET AOD as ground-truth, the diagnostic EE envelope includes all possible sources of uncertainties due to for example, changes of the card type, burning threshold, cloud contamination, changes in aerosol properties during the day and uncertainties in sunshine duration measurements. To derive the EE estimates, we use a random subset of about 7000 AOD retrievals from the validation dataset as described previously with all stations. We divide the data into 100 bins, so that each bin contains the same amount of measurements. We compute the standard deviation of the retrieval error and the average retrieved AOD in each bin. We intentionally select the retrieved AOD as the variable to compare the uncertainty against so that it is possible to estimate the retrieval uncertainties also in cases in which accurate or measured AOD is not available. We fit a linear model to the uncertainty data and derive the EE envelope estimate as follows:

$$EE_{AOD} = \pm(0.01 + 0.40 \times AOD_{NHM}) \qquad\qquad (8)$$

where $AOD_{NHM}$ is the retrieved AOD from the NHM method."

Sayer, A. M., Govaerts, Y., Kolmonen, P., Lipponen, A., Luffarelli, M., Mielonen, T., Patadia, F., Popp, T., Povey, A. C., Stebel, K., and Witek, M. L.: A review and framework for the evaluation of pixel-level uncertainty estimates in satellite aerosol remote sensing, Atmos. Meas. Tech., 13, 373–404, https://doi.org/10.5194/amt-13-373-2020, 2020.

Levy, R. C., Remer, L. A., Kleidman, R. G., Mattoo, S., Ichoku, C., Kahn, R., and Eck, T. F.: Global evaluation of the Collection 5 MODIS dark-target aerosol products over land, *Atmos. Chem. Phys.*, 10, 10399–10420, doi: 10.5194/acp-10-10399-2010, 2010.

Minor comments

*Comment 1. Page2 L15 AERONET has the denser network but looking at the site I can see only four stations with data series more than 15 years. Probably WMO-GAW network or Skynet network can be mentioned too.*

Thanks. We have mentioned both WMO-GAW and Skynet networks in the relevant text as follows:

"It has been mainly measured using reference instruments, sun photometers for instance, from various ground-based networks. Among them are the Aerosol Robotic Network (AERONET; Holben et al., 1998), the Global Atmospheric Watch Precision Filter Radiometer network (McArthur et al., 2003) and the SKYradiometer NETwork (Aoki et al., 2006)."

*Comment 2. 15 line 15 Ångström*

Thanks. Done as requested in the whole manuscript.

*Comment 3. Why is the effective wavelength in BAOD close to AOD at 750 nm?*

Thanks. Based on our analysis, we acknowledge there is a variability in the effective wavelength of BAOD depending on the atmospheric state. However, we found the assumed 750 nm as effective wavelength in BAOD as a reasonable compromise knowing that the true

atmospheric state is not always known, especially during the reconstruction period. In addition, we have already mentioned in the manuscript that other studies (Qiu, 1998; Molineaux et al., 1998) have reported similar assumptions.

Molineaux, B., Ineichen, P., O'Neil, N.: Equivalence of pyrheliometric and monochromatic aerosol optical depths at a single key wavelength, *Appl. Opt.*, 37(30), 7008–7018, doi: 10.1364/AO.37.007008, 1998.

Qiu, J. H.: A method to determine atmospheric aerosol optical depth using total direct solar radiation, *J. Atmos. Sci.*, 55, 744–757, doi:10.1175/1520-0469, 1998.

*Comment 4. How can anyone trust Merra data for small to moderate aerosol changes in the past when Pinatubo and El Chichon are not visible in the dataset?*

Thanks. We want to stress that in the MERRA data, volcanic sources are included in the aerosol assimilation process. In addition, it is clearly shown in the time series of monthly means over the major aerosol source regions as illustrated in the Figure 13 of Gelaro et al., 2017 or in the winter series of Figure 6 of our manuscript, where the El Chichon and Pinatubo events are visible in the time series. Nevertheless, it is worth mentioning that we have already pointed some limitations of the MERRA AOD dataset in the manuscript in Section 6.3.

Gelaro, R., McCarty, W., Suárez, M. J., Todling, R., Molod, A., Takacs, L., Randles, C. A., Darmenov, A., Bosilovich, M.G., Reichle, R. and Wargan, K.: The modern-era retrospective analysis for research and applications, version 2 (MERRA-2), *J. Clim.*, *30*(14), 5419-5454, doi: 10.1175/JCLI-D-16-0758.1, 2017.

---

## Author Comment (AC2) · 31 Mar 2020

**ANSWERS TO REFEREE #2**

First of all, we thank Referee #2 for these positive constructive remarks and comments. The comments have been addressed below and have been taken into account for revising the manuscript. The responses are below after the reviewer points that are in italics.

**Major comments**

In this work, a new method for estimating AOD from SD measurements under cloud-free days is proposed. The adopted method relies on a broadband DNI model that takes into account the local conditions affecting SD measurements. The work is well suited to the readership of AMT. However, there are a number of issues with the procedure developed in the paper that prevent its publication in the current form. The main issues with the work concern the decision to use daily mean of cloud cover data obtained only from four observations terms (every 6-hours). There are also other meteorological databases which contain a variety of hourly meteorological data from weather stations located in Europe (e.g. https://www.ncdc.noaa.gov/isd). Therefore, the authors must quantify if there are significant differences between selecting the cloud-free days using hourly cloud data and ECAD dataset. Also, satellite cloud cover product data could be used in the analysis, as they provide hourly cloud for than cover data more twenty years (https://www.cmsaf.eu/EN/Products/AvailableProducts/Climate\_Data\_records/Climate\_Data \_*Records\_node.html*)

Thank you for this valuable remark and the suggested database i.e. Integrated Surface Database (ISD) for hourly meteorological data. It will be useful for the subsequent work which aims to apply the proposed method for the reconstruction of the historical evolution of aerosol load at the global scale with the best possible ground-based measurements, while here we presented, validated and applied the method for a couple of stations. Nevertheless, we have explored the ISD database with a focus on cloud information. For those ground-based stations where SD measurements are also available, cloud cover measurements are mostly provided with 6-hourly temporal resolution in the past i.e. before 1980 (and it is worth remembering that our major goal is to reconstruct past AOD before the satellite era). This 6-hourly cloud cover

measurements are still (up to now) given for some stations. In general, this temporal resolution is the most widely available resolution for numerous stations over the world within the reconstruction period.

In this study, a cloud-free day is assumed when the mean daily value of TCC is rounded to 0 okta. That average value is computed from three-daily observations, typically at 06:00, 12:00 and 18:00 UTC. Obviously, it is a very low temporal resolution for cloud-free days selection because within a 6-hour period, the sky can be absolutely obscured by clouds resulting in an erroneous selection. Therefore, and following the referee comment, it is relevant to quantify how accurate the selection algorithm is when using 6-hourly observations instead of a much higher temporal resolution such as 1-hourly observations as requested by the referee. In other words, which proportion of days with cloud influence in SD is included in the set of clear sky days based on our selection algorithm.

In order to achieve this goal, we selected the well-known KNMI database instead of the ISD database because of the availability of metadata. This latter allows avoiding as much as possible inhomogeneity problems caused by station movements and changes in instruments. The KNMI database consists of hourly observations of numerous meteorological parameters from about 30 ground-based stations. The automatized hourly cloud cover measurements can be downloaded through the website http://projects.knmi.nl/klimatologie/uurgegevens/selectie.cgi. We used the temporal coverage from 2010 onwards. For the sake of clarity, let TCC1 and TCC6 denote the daily averages computed from 1-hourly and 6-hourly data, respectively. According to the WMO guidelines, TCC of 0 okta corresponds to completely cloud-free sky, so it was considered that the limit of 1 okta could be extended to represent almost cloud-free sky conditions representing cases when clouds do not obstruct the SD measurements.

For each station, we retained days obeying to these following filters: (1) no missing hourly TCC between sunrise and sunset, (2) no missing TCC values at 06:00, 12:00 and 18:00 UTC and (3) rounded TCC6 is equal to 0. Therefore, we investigated the set of remaining days. Figure 1 shows the distribution of frequency of days over the Netherlands. The x-axis is the okta bin with a width of 0.5 based on TCC1 represented by the upper limit and the y-axis is the maximum value from the 1-hourly cloud observations between sunrise and sunset.

Figure 1. Relative frequency of days per okta bin with respect to the maximum cloud cover measurements between the sunrise and sunset.

Regarding the TCC1 threshold of 1 okta for almost cloud-free days, one can observe that the selection algorithm successfully captures cloud-free days in 87% of cases. Only 13% of the cases have a significant cloud influence. That demonstrates the good performance of the selection algorithm. For these 13% of cases, the retrieved AOD are overestimated by cloud contamination whose the impact is considered in the uncertainty estimates. For more detailed information on the uncertainty estimates, please refer to the answer to first comment of Referee #1.

Minor issues:

*Comment 1. Page 5 Line 20. The authors must mention the studies in which was performed the comparison between ECMWF reanalysis product and AERONET data.*

Thanks. The comparison was carried out by ourselves and we have re-written the relevant part accordingly as follows:

"A comparison between ECWMF and AERONET daily TWC is carried out over Europe with AERONET measurements serving as reference. On average, we found that the ECMWF values slightly underestimate TWC by 4%, or below 1 kg m-2 with a correlation coefficient of 0.8 and a small standard deviation"

*Comment 2. Page 6 Line 5. Which are the studies that demonstrate the performances of TOC estimates?*

Thanks. The answer is the same as for the previous comment. We have re-written the relevant part accordingly as follows:

"Our comparisons of TOC from ECMWF and OMI serving as reference show on average a high correlation of 0.9, a low bias of +3% and very limited spread of points denoting the accuracy of ECMWF TOC."

Comment 3. Page 6 Line 15. The authors must motivate why they have used the MxD08\_D3mModis product instead of the higher spatial and temporal resolution Modis aerosols data (MxD04\_L2product).

Thank you very much for this remark. We have added the motivation in the relevant part of the paragraph as follows:

"Despite of the coarser spatial resolution than the corresponding one for the level-2 product, the level-3 product was used because it takes into account all overpasses of the instrument during the same day, thereby making it to be more representative of daily average aerosol properties than the level-2 instantaneous data products."

Comment 4. Page 10 Line 20. For which station the 12000 the clear sky atmospheric states were built?

Thanks. No specific station was needed here. The set of 12 000 clear sky atmospheric states was built for comparisons between the robust model and the simplified model with necessary inputs describing the state of the atmosphere.

Comment 5. Page 15 Line 5. The authors must discuss why most of the stations selected after applying the homogenization test are located in the Mediterranean climate. It is hard to believe

that, excepting data from two stations, only the stations located in Spain provide homogenous SDF data.

Thanks. The reviewer is right as most of the selected stations in Table are from Spain It is worth mentioning that Spain is one of the countries with more long-term measurements of SD and TCC available on ECA&D, in contrast with other countries such as United Kingdom or, France where data is not provided. We have slightly modified the relevant part of the text to make it clear as follows:

"To ensure a reasonable set of ground-based stations as well as a good quality of measurements, four criteria have been applied on those ECA&D stations having both SD and TCC measurements within a calendar day. First, to include at least the brightening period, ECA&D stations having SD measurements starting before the year 1985 were chosen. Second, ECA&D stations should be collocated with AERONET stations with a maximum distance of 50 km. Third, the overlapping period where that both cloud-free SD and AERONET measurements are available should cover at least three years. Finally, as the fourth criteria, temporal homogeneity tests have been applied on SD measurements to select stations with homogeneous time series. For this last one, three main tests are used:"

*Comment 6. Page 18 Line 10. The authors must provide the burning thresholds computed for all the stations as an ESM table. This will add value to the paper.*

Thanks. We have added an ESM Table as requested and have written in the text as follows:

"See Table S1 in the supplement for computed monthly burning thresholds for all stations."

Comment 7. Page 19 Line 5. In order to better emphasize the results of the three models tested, trend lines for each model should be added on the scatterplot.

Thanks. Done as requested.

Comment 8. Some references mentioned in the text are missing from the Bibliography section (e.g. Qui, 1998).

Thanks. We have carefully checked the Bibliography accordingly to citations in the text.

---

## Author Response (AR2)

ANSWERS TO REFEREE #1

We thank Referee #1 for these positive remarks and comments. The comments have been addressed below and have been taken into account for revising the manuscript. The responses are below after the reviewer points that are in italics.

*In my opinion the authors have improved the manuscript based on the recommendations. Concerning my initial review comments, I just would like to mention:*

*Comment 1- Authors dealing with the uncertainty analysis would be complete if you could add the figure from which the (new) equation 8 has been extracted.*

Thanks. Done as requested.

*Comment 2- concerning cloud cover, the 3 points per day availability is the best existing. So I understand the limitations. However, it is quite simple to think that during all days with 0 cloud cover over these 3 measurements during the day there will be a number of days with some (e.g. cirrus) clouds in between. That will systematically bias the AOD retrieval to higher values. I guess authors just have to mention this possibility?*

Thank you for this remark. Since both referees found the analysis we carried out on this issue very interesting, and in order to help AMT readers to clearly understand it, we have included in the manuscript a new sub-section Section 6.1 (6-hourly *vs* 1-hourly cloud cover data for the selection of cloud-free days) where we mentioned the possibility as requested by referee #1 in more details.

ANSWERS TO REFEREE #2

We thank Referee #2 for these positive remarks and comments. The comments have been addressed below and have been taken into account for revising the manuscript. The responses are below after the reviewer points that are in italics.

*Comment-1. It is not clear if the authors investigated the possibility of using remote sensing cloud cover products in the study. The authors must mention in the manuscript if it is feasible or not to use this type of data in the reconstruction of AOD.*

Thank you for this remark. We have mentioned the possibility in the relevant part of the text as follows:

"Another way for cloud screening would be to use remote sensing cloud cover products. Unfortunately, this type of data suffers from a lack of temporal availability, especially before the 1980s, and their spatial resolution is neither good enough. This is why using ground-based measurements of total cloud cover is still the most reasonable way for the selection of cloud-free days for the scope of this study."

*Comment-2. Although the authors mention in the answer to my main comment that: "In this study, a cloud-free day is assumed when the mean daily value of TCC is rounded to 0 okta. That average value is computed from three-daily observations, typically at 06:00, 12:00 and 18:00 UTC.", the ATB of the ECAD daily data reveals that "Daily mean cloud cover CC: Whenever synoptical cloud cover data is available at 00, 06, 12 and/or 18 UT, mean daily cloud cover is calculated as the average of the available values" (in document https://eca.knmi.nl//documents/atbd.pdf, page 9). The authors must clarify this aspect.*

Thanks for this remark. After checking the meta-information in more detail for each station used, we have included a clarification in the relevant part of the text as follows:

"For each station selected for this study, daily means are obtained from the average of at least 3 observations per day taken at around 06:00, 12:00 and 18:00 UT".

*Comment-3. The authors must verify the link which points to the ECAD dataset (Section 2.2); the link assigned belongs to the Finish Met. Institute.*

Thanks. Done as requested.

*Comment-4. Apparently, there is a typo in the caption of figure 5.*

Thanks. Done as requested.